# The Discrete and Continuous Retardation and Relaxation Spectrum Method for Viscoelastic Characterization of Warm Mix Crumb Rubber-Modified Asphalt Mixtures

**DOI:** 10.3390/ma13173723

**Published:** 2020-08-23

**Authors:** Fei Zhang, Lan Wang, Chao Li, Yongming Xing

**Affiliations:** 1School of Science, Inner Mongolia University of Technology, Hohhot 010040, China; cx15754714810@tom.com; 2School of Civil Engineering, Inner Mongolia University of Technology, Hohhot 010040, China; wl8802616@aliyun.com (L.W.); nmggydx@21cn.com (C.L.)

**Keywords:** master curve, discrete spectrum, continuous spectrum, relaxation modulus, creep compliance, warm mixing crumb rubber-modified asphalt mixture

## Abstract

To study the linear viscoelastic (LVE) of crumb rubber-modified asphalt mixtures before and after the warm mix additive was added methods of obtaining the discrete and continuous spectrum are presented. Besides, the relaxation modulus and creep compliance are constructed from the discrete and continuous spectrum, respectively. The discrete spectrum of asphalt mixtures can be obtained from dynamic modulus test results according to the generalized Maxwell model (GMM) and the generalized Kelvin model (GKM). Similarly, the continuous spectrum of asphalt mixtures can be obtained from the dynamic modulus test data via the inverse integral transformation. In this paper, the test procedure for all specimens was ensured to be completed in the LVE range. The results show that the discrete spectrum and the continuous spectrum have similar shapes, but the magnitude and position of the spectrum peaks is different. The continuous spectrum can be considered as the limiting case of the discrete spectrum. The relaxation modulus and creep compliance constructed by the discrete and continuous spectrum are almost indistinguishable in the reduced time range of 10^−5^ s–10^3^ s. However, there are more significant errors outside the time range, and the maximum error is up to 55%.

## 1. Introduction

It is well known that asphalt mixtures demonstrate linear viscoelastic (LVE) solid properties at small strain levels (typically no more than 120 uε) and over a wide range of frequencies and temperatures [1]. These viscoelastic response functions, including dynamic modulus |E*(f)|, creep compliance D(t), and relaxation modulus E(t), are used to characterize the LVE properties of hot mix asphalt (HMA). The dynamic modulus is the ratio of the amplitude of the stress and strain response of viscoelastic material to endure dynamic loading, which can be used to determine the strength properties of viscoelastic materials. Creep compliance represents the time-dependent strain response of viscoelastic material in unit stress, which can be used to determine the viscoelastic strain. Conversely, the relaxation modulus represents the time-dependent stress response of viscoelastic material in unit strain, which can be used to determine the viscoelastic stress. Different viscoelastic response functions characterize the viscoelasticity of asphalt mixtures in different forms. Different experimental tests can characterize the corresponding viscoelastic response functions, respectively. However, it is theoretically feasible to convert these LVE functions to each other, since they are mathematically equivalent for each loading mode [2].

The concept of spectrum distribution function [3] plays a significant role in LVE behavior theory. It is independent of a specific time system and occupies a vital position in viscoelasticity theory. The relaxation spectrum and the retardation spectrum are the distributions of the spectrum intensity with respect to relaxation and retardation time. They are an essential characteristic of LVE materials, from which all other viscoelastic functions and their responses can be obtained [4]. These functions describe the primary response of viscoelastic materials, so it has great significance to determine and characterize them from the given experimental results. Generally, they can be divided into the continuous spectrum and discrete spectrum according to the time spectrum interval [5].

The discrete relaxation and retardation spectrum of asphalt mixtures can be characterized by mechanical models with different configurations of linear springs and dashpots [6]. The discrete relaxation spectrum can be determined by Prony coefficients from the generalized Maxwell model (GMM) model. In contrast, the Prony coefficients of the generalized Kelvin model (GKM) can be used to determine the discrete retardation spectrum [7]. A series of methods for determining the discrete spectrum from experimental data have been proposed. However, oscillations and negative spectrum intensities are unavoidable problems with the discrete spectrum due to the scattered nature of the experimental data and the time interval choice. Schapery [8] uses the collocation method to solve the discrete spectrum. In a word, using the data of the pre-selected collocation points to solve the corresponding spectrum intensity, but this is an ill-posed problem, and the solution result is not unique. The accuracy of the spectrum intensity depends on the spacing of the collocation points [9]. However, the collocation method is susceptible to the value of the discrete-time interval. To address the problems with the collocation method, Cost and Becker [10] proposed the multiple data method by applying the least-squares fitting technique to the Laplace transform domain. The recursive method proposed by Emri and Tschoegl [11,12,13] further improves the calculation method of the discrete spectrum. Park and Kim [14] proposed a method of pre-smoothing experimental data by power-law sequence representation and then fitting the Prony series to the pre-smoothed data. This method avoids the occurrence of negative spectrum intensity and local oscillation of the spectrum. Besides, the regularization method of quadratic programming [15], Marquardt-Levenberg program [16], Bayesian method [17] and piecewise cubic Hermite spline polynomials [18] are new methods that have been gradually developed in recent years. The discrete relaxation spectrum is an essential parameter for predicting the nonlinear viscoelastic behavior, while the continuous spectrum is not convenient for numerical calculations. Therefore, Bae [19] developed a new method to detect the traditional discrete relaxation spectrum by continuous relaxation spectrum, then allows precise determination of the discrete relaxation spectrum.

Considering the problems with discrete spectrum, the continuous spectrum concept has been gradually developed in recent years. The continuous spectrum can be considered as a discrete spectrum with an infinitely dense time interval [20]. Using the continuous spectrum instead of the discrete spectrum can avoid the above problems. Although we can, obtain the continuous spectrum from a viscoelastic function requires integral inversion, which usually presents a considerable challenge. Bažant and Xi [21] obtained the continuous retardation spectrum using the inverse Laplace transform and the higher-order derivative of the creep compliance function. Mun and Zi [22] adopted a similar method, converting the frequency-domain test results into time-domain test results using an approximate method, and constructing the master curve in the time domain, and then using the master curve to obtain higher-order derivatives and finally obtain the continuous retardation spectrum. Based on the LVE theory, Sun [23] constructed continuous relaxation and retardation spectrum based on the H-N complex modulus model and compared it with the results of the discrete spectrum. Liu [24]. proposed a new method for determining the continuous relaxation spectrum. This method solves the continuous relaxation spectrum simultaneously from the master curve expressions of the storage modulus E′ and the loss modulus E″.

Although the computational methods of discrete and continuous spectrum have achieved an unprecedented development, more and more people are paying great attention to the interconversion between each other [25]. There are some results comparing the viscoelastic functions constructed from the discrete and continuous spectrum. Gu [26] used the discrete spectrum of the 2S2P1D model to obtain the viscoelastic information of a fine aggregate matrix to construct discrete element models. Yu [27] used the generalized sigmoidal model (GSM) and Havriliak-Negami model (HNM) to construct master curve models and then compared the differences between the discrete and continuous spectra constructed by the different models. Relaxation modulus and creep compliance can be derived from relaxation and retardation spectra, respectively. Teltayev [28] presented the results of bending beam rheometer (BBR) tests of blown bitumen at low temperatures and analyzed the results using the developed relaxation modulus and relaxation time spectra. The framework developed by Liu [29] based on spectral methods ensures that the master curves of all viscoelastic variables complied with the LVE theory. Moreover, the relaxation and retardation spectrum are the most important parameters for the construction of viscoelastic continuous damage models (VECD) [30,31,32] and their probabilistic models [33,34].

This study focused on comparing the relaxation modulus and creep compliance master curves of crumb rubber-modified asphalt mixtures constructed by discrete and continuous spectrum methods and to evaluate the accuracy of the master curve model. In this paper, four types of crumb rubber-modified asphalt mixtures were produced in laboratory, under the premise of approximately satisfying the Kramers-Kronig (K-K) relations [35], the coefficients of the Prony series can be treated as a discrete spectrum according to the collection method. The inverse integral transformation [36] can be used to derive the continuous spectrum according to the master curve model of generalized Sigmoidal. Finally, the viscoelastic response functions constructed based on the discrete and continuous spectrum are compared, and the error magnitudes were analyzed.

## 2. Materials and Methods

### 2.1. Materials and Specimen Fabrication

Four types of laboratory-produced mixtures were used in this study. AC-16, which is also used widely in highway construction in China, was utilized in the following text. The virgin asphalt PG 64–22 was purchased from Inner Mongolia Luda Asphalt Co., Ltd. (Jining, China). The mineral aggregate was purchased from a quarry located in Zhuozishan (Inner Mongolia Autonomous Region, China). The warm mix additive (SDYK) was purchased from MeadWestvaco Co., Ltd. (Wuxi, China). Crumb rubber was purchased from the Hubei Anjie Road & Bridge Technology Co., Ltd. (Ezhou, China). Table 1 and Figure 1 show the aggregate gradation of mixtures. The crumb rubber-modified asphalt binder mixing method used in this study was a wet process, in which the crumb rubber is added to the virgin asphalt binder (penetration grade 80/100) before introducing it into the asphalt mixture. The crumb rubber-modified asphalt binder was produced in the laboratory at 180 °C for 30 min using an open blade mixer at a blending speed of 700 rpm [37]. The percentage of crumb rubber added for the crumb rubber-modified asphalt binder was 20% by weight of virgin asphalt. For the warm mix crumb rubber- modified asphalt binder, the warm mix additive was added to crumb rubber-modified asphalt binder by mixing at 180 °C for 30 min with a conventional mechanical mixer. The storage stability of binder can be characterized by the difference in softening points, the results are shown in Table 2. Then the crumb rubber-modified asphalt binder was prepared for manufacturing specimens.

Hot mix 60-mesh crumb rubber-modified asphalt mixture (HMA-60) is the mixture prepared by mixing aggregates and 60-mesh crumb rubber-modified asphalt binder following the hot mixing process, while warm mix 60-mesh crumb rubber-modified asphalt mixture (WMA-60) is the mixture prepared by mixing aggregate and warm mix 60-mesh crumb rubber-modified asphalt binder following the warm mixing process. Hot mix compound-mesh crumb rubber-modified asphalt mixture (HMA-C) is the mixture prepared by mixing aggregates and compound-mesh crumb rubber- modified asphalt binder following the hot mixing process. Warm mix compound-mesh crumb rubber-modified asphalt mixture (WMA-C) is the mixture prepared by mixing aggregates and warm mix compound-mesh crumb rubber-modified asphalt binder following the warm mixing process.

The crumb rubber-modified asphalt content is 5.4% and 5.6% by the weight of the total mixture mass for HMA-60 and HMA-C, respectively. The content of crumb rubber-modified asphalt of warm mix asphalt mixture is the same with the corresponding hot mix asphalt mixture. HMA was mixed and compacted at 180 °C and 170 °C, respectively, while WMA was mixed and compacted at 162 °C and 152 °C, respectively. The volume parameters of warm-mixed asphalt mixture were consistent with the corresponding hot-mixed mixture. Table 3 lists the volume parameters of the mixture. The raw specimens (150 mm in diameter and 178 mm in height) were fabricated using a Superpave gyratory compactor (produced by IPC Company, Melbourne, Australia) and then cut and cored to standard dimensions (100 mm in diameter and 150 mm in height). The air void content of all type mixture was 4 ± 0.5%. Three replicate specimens were fabricated and tested for every mixture type. Prior to testing all specimens were stored in an unlit cabinet to reduce ageing, but not allowed to more than two weeks.

### 2.2. Experimental Plan

The complex modulus test was conducted in a compression load according to AASHTO TP 79–15 [38]. A servo-hydraulic universal testing machine (UTM-100) produced by the IPC Company (Melbourne, Australia) was used to measure dynamic modulus and phase angle for all test specimens. The test was conducted in four different temperature (5 °C, 20 °C, 35 °C, 50 °C) and seven different frequency (25 Hz, 20 Hz, 10 Hz, 5 Hz, 1 Hz, 0.5 Hz, 0.1 Hz). The test started from the lowest temperature (5 °C) to the highest (50 °C) for each specimen. Moreover, each specimen was tested at every temperature started from the highest frequency (25 Hz) to the lowest (0.1 Hz). The maximum axial strain was controlled within 70 με to assure that the specimens stayed within the LVE domain [38]. The raw data on dynamic modulus and phase angle for types of mixture is shown in Appendix A. The information on the statistical analysis of the test results of the types of asphalt mixture is shown in Table A1 and Table A2.

## 3. Theoretical Background

### 3.1. LVE Theory Asphalt Mixtures

#### 3.1.1. The Time-Temperature Superposition Principle (TTSP)

The viscoelastic of asphalt mixtures is significantly dependent on time and temperature. In polymer physics, the TTSP [39] can be used to analyze material properties in unknown conditions based on known conditions. Then, the master curve of viscoelastic function to the reduced frequency is obtained, which would greatly reduce the test workload. the reduced frequency can be calculated following Equations (1) and (2).

Williams-Landel-Ferry [40] applied the TTSP to polymers and found an empirical expression for the shift factor. It is the WLF equation. It has been found that the WLF equation can fit the observed shift factor data over a wide range of temperatures from T_g_~T_g_ + 100 for asphalt binders and mixtures, the individual expression is shown in Equation (1).
(1)$$\mathrm{lg}\alpha_{T}=\frac{-C_{1}\left(T-T_{r}\right)}{C_{2}+\left(T-T_{r}\right)}$$
where: $lg\alpha_{T}$ is the shift factor; *C*_1_, *C*_2_ are positive test constants of WLF for the material; *T_r_* and *T* are the reference temperature and the experimental temperature respectively, °C. in this paper, the reference temperature *T_r_* was taken as 20 °C.

Reduced frequency [22] *f_r_* is the equivalent frequency of experimental temperature to the reference temperature, and it can be calculated as Equation (2) in the logarithmic axis:(2)$$\mathrm{lg}f_{r}=\mathrm{lg}f+\mathrm{lg}\alpha_{T}$$
where: $lgf_{r}$ is the reduced frequency in reference temperature; lgf is the frequency in experiment temperature.

#### 3.1.2. The Method of Construction the Master Curve of Linear Viscoelastic Response Function

The generalized sigmoidal model (GSM) [41,42] and Huet-Sayegh model (HSM) [43,44] have been used successfully to characterize the mechanical behavior of asphalt mixtures, and this study uses the model to construct the dynamic modulus, storage modulus, and storage compliance master curves of asphalt mixtures. Based on the K-K relations, the phase angle, loss modulus, and loss compliance master curves can be derived from the GSM. The master curves constructed using this method, including all the viscoelastic information. whose expression is presented as follows.

##### The Master Curve of Dynamic Modulus and Phase Angle

The master curve of dynamic modulus can be presented by GSM. The master curve of phase angle can be derived from GSM following the approximate K-K relations. The error function $ef_{1}$ was applied to the test data of dynamic modulus and phase. Its equation is shown in Equation (3). The master curve model parameters in Equations (4) and (5) for each mixture type were solved simultaneously by minimizing the error function $ef_{1}$ (Equation (3)).
(3)$$ef_{1}=ef_{E^{*}}+ef_{\phi }=\frac{1}{N}\sqrt{\overset{N}{\underset{i=1}{\sum }}{\left(\frac{{\left|E^{*}\right|}_{m,i}-{\left|E^{*}\right|}_{p,i}}{{\left|E^{*}\right|}_{m,i}}\right)}^{2}}+\frac{1}{N}\sqrt{\overset{N}{\underset{i=1}{\sum }}{\left(\frac{{\left|\phi \right|}_{m,i}-{\left|\phi \right|}_{p,i}}{{\left|\phi \right|}_{m,i}}\right)}^{2}}$$
where: $ef_{1}$ is the error function of dynamic modulus and phase angle; $ef_{E^{*}}$ is the error function of dynamic modulus; $ef_{\Phi }$ is the error function of phase angle; N is the number of the measured data points that is equal to 28; ${\left|E^{*}\right|}_{m,i}$ is ith data point of the measured dynamic modulus, MPa; ${\left|E^{*}\right|}_{p,i}$ is ith data point of predicted by dynamic modulus master curve, MPa; ${\left|\phi \right|}_{m,i}$ is ith data point of the measured phase angle, °; ${\left|\phi \right|}_{p,i}$ is ith data point of predicted by phase angle master curve, °.
(4)$$\mathrm{lg}\left|E^{*}\right|=\delta +\frac{\alpha }{1+e^{\beta +\gamma \left(\log f_{r}\right)}}$$
(5)$$\phi_{1}\left(f_{r}\right)=k\frac{\pi }{2}\frac{d\left(\mathrm{lg}\left|E^{*}\right|\right)}{d\left(\mathrm{lg}f_{r}\right)}=-\frac{\pi }{2}k\alpha \gamma \frac{e^{\beta +\gamma \mathrm{lg}f_{r}}}{{\left(1+\lambda e^{\beta +\gamma \mathrm{lg}f_{r}}\right)}^{1+\frac{1}{\lambda }}}$$
where: $lg\left|E^{*}\right|$ is the logarithm of the dynamic modulus; δ is the value of the lower asymptote of the $\left|E^{*}\right|$ master curve; α is the difference between the upper and lower asymptotes of the $\left|E^{*}\right|$ master curve; β and γ is shape coefficients of the $\left|E^{*}\right|$ master curve;. λ determined the model’s asymmetric characteristics; $\Phi_{1}\left(f_{r}\right)$ the master curve of phase angle predicted by the GSM based on the approximate K-K relations; k is a positive correction, it is to obtain potentially more accurate prediction.

##### The Master Curve of Storage Modulus and Loss Modulus

The master curve of storage modulus can be presented by GSM. The master curve of the loss modulus can be derived from GSM following the approximate K-K relations. The error function $ef_{2}$ was applied to the test data of storage modulus and loss modulus. The equation is shown in Equation (6). The test data of storage modulus and loss modulus can be determined from the results of the complex modulus test following Equations (7) and (8). The master curve model parameters in Equations (9) and (10) for each mixture type were solved simultaneously by minimizing the error function $ef_{2}$ (Equation(6)).
(6)$$ef_{2}=ef_{E^{\prime }}+ef_{E^{\prime \prime }}=\frac{1}{N}\sqrt{\overset{N}{\underset{i=1}{\sum }}{\left(\frac{{\left|E^{\prime }\right|}_{m,i}-{\left|E^{\prime }\right|}_{p,i}}{{\left|E^{\prime }\right|}_{m,i}}\right)}^{2}}+\frac{1}{N}\sqrt{\overset{N}{\underset{i=1}{\sum }}{\left(\frac{{\left|E^{\prime \prime }\right|}_{m,i}-{\left|E^{\prime \prime }\right|}_{p,i}}{{\left|E^{\prime \prime }\right|}_{m,i}}\right)}^{2}}$$
where: $ef_{2}$ is the error function of storage modulus and loss modulus; $ef_{E^{\prime }}$ is the error function of storage modulus; $ef_{E^{\prime \prime }}$ is the error function of loss modulus; N is the number of the measured data points that is equal to 28; ${\left|E^{\prime }\right|}_{m,i}$ is ith data point of the measured storage modulus, MPa; ${\left|E^{\prime }\right|}_{p,i}$ is ith data point predicted by storage modulus master curve, MPa; ${\left|E^{\prime \prime }\right|}_{m,i}$ is ith data point of the measured loss modulus, MPa; ${\left|E^{\prime \prime }\right|}_{p,i}$ is ith data point predicted by loss modulus master curve, MPa.
(7)$$E^{\prime }=\left|E^{*}\right|\cos\left(\phi \right)$$
(8)$$E^{\prime \prime }=\left|E^{*}\right|\sin\left(\phi \right)$$
where: $E^{\prime }$ is the storage modulus obtained from complex modulus test; $E^{\prime \prime }$ is the loss modulus obtained from complex modulus test; Φ is the phase angle obtained from complex modulus test:(9)$$\mathrm{lg}E^{\prime }=\delta^{\prime }+\frac{\alpha^{\prime }}{{\left(1+\lambda^{\prime }e^{\beta^{\prime }+\gamma^{\prime }\left(\log f_{r}\right)}\right)}^{\frac{1}{\lambda^{\prime }}}}$$
(10)$$E^{\prime \prime }=\frac{\pi }{2}kE^{\prime }\frac{d\left(logE^{\prime }\right)}{d\left(\log f_{r}\right)}=\frac{\pi }{2}k^{\prime }E^{\prime }\alpha^{\prime }\gamma^{\prime }\frac{e^{\beta^{\prime }+\gamma^{\prime }\mathrm{lg}f_{r}}}{{\left(1+\lambda^{\prime }e^{\beta^{\prime }+\gamma^{\prime }\mathrm{lg}f_{r}}\right)}^{1+\frac{1}{\lambda^{\prime }}}}$$
where: $lgE^{\prime }$ is the logarithm of the storage modulus; $\delta^{\prime }$ is the value of the lower asymptote of the $E^{\prime }$ master curve; $\alpha^{\prime }$ is the difference between the upper and lower asymptotes of the $E^{\prime }$ master curve; $\beta^{\prime }$ and $\gamma^{\prime }$ is shape coefficients of the $E^{\prime }$ master curve;. $\lambda^{\prime }$ determined the model’s asymmetric characteristics; $E^{\prime \prime }$ is the master curve of loss modulus predicted by the GSM based on the approximate K-K relations; $k^{\prime }$ is a positive correction, used to obtain potentially more accurate predictions.

##### The Master Curve of Storage Compliance and Loss Compliance

The master curve of storage compliance can be presented by GSM. The master curve of the loss compliance can be derived from GSM following the approximate K-K relations. The error function $ef_{3}$ was applied to the test data of storage compliance and loss compliance. The equation is shown in Equation (11). The test data of storage compliance and loss compliance can be determined from the complex modulus test results following Equation (12). The master curve model parameters in Equations (13) and (14) for each mixture type was solved simultaneously by minimizing the error function $ef_{3}$ (Equation(11)).
(11)$$ef_{3}=ef_{D^{\prime }}+ef_{D^{\prime \prime }}=\frac{1}{N}\sqrt{\overset{N}{\underset{i=1}{\sum }}{\left(\frac{{\left|D^{\prime }\right|}_{m,i}-{\left|D^{\prime }\right|}_{p,i}}{{\left|D^{\prime }\right|}_{m,i}}\right)}^{2}}+\frac{1}{N}\sqrt{\overset{N}{\underset{i=1}{\sum }}{\left(\frac{{\left|D^{\prime \prime }\right|}_{m,i}-{\left|D^{\prime \prime }\right|}_{p,i}}{{\left|D^{\prime \prime }\right|}_{m,i}}\right)}^{2}}$$
where: $ef_{3}$ is the error function of storage compliance and loss compliance; $ef_{D^{\prime }}$ is the error function of storage compliance; $ef_{D^{\prime \prime }}$ is the error function of loss compliance; N is the number of the measured data points that is equal to 28; ${\left|D^{\prime }\right|}_{m,i}$ is the *i*-th data point of the measured storage compliance, MPa^−1^; ${\left|D^{\prime }\right|}_{p,i}$ is *i*-th data point predicted by the storage compliance master curve, MPa^−1^; ${\left|D^{\prime \prime }\right|}_{m,i}$ is *i*-th data point of the measured loss compliance, MPa^−1^; ${\left|D^{\prime \prime }\right|}_{p,i}$ is ith data point predicted by the loss compliance master curve, MPa^−1^.
(12)$$D^{*}=\frac{1}{E^{*}}=D^{\prime }-iD^{\prime \prime }=\frac{E^{\prime }}{E^{\prime 2}+E^{\prime \prime 2}}-i\frac{E^{\prime \prime }}{E^{\prime 2}+E^{\prime \prime 2}}$$
(13)$$\mathrm{lg}\left|D^{\prime }\right|=\delta^{\prime \prime }-\frac{\alpha^{\prime \prime }}{{\left(1+\lambda^{\prime \prime }e^{\beta^{\prime \prime }-\gamma^{\prime \prime }\left(\log f_{r}\right)}\right)}^{\frac{1}{\lambda^{\prime \prime }}}}$$
(14)$$D^{\prime \prime }=-\frac{\pi }{2}k^{\prime \prime }D^{\prime }\frac{d\left(logE^{\prime }\right)}{d\left(\log f_{r}\right)}=\frac{\pi }{2}k^{\prime \prime }D^{\prime }\alpha^{\prime \prime }\gamma^{\prime \prime }\frac{e^{\beta^{\prime \prime }-\gamma^{\prime \prime }\mathrm{lg}f_{r}}}{{\left(1+\lambda^{\prime \prime }e^{\beta^{\prime \prime }-\gamma^{\prime \prime }\mathrm{lg}f_{r}}\right)}^{1+\frac{1}{\lambda^{\prime \prime }}}}$$
where: $D^{*}$ is the complex compliance; $E^{*}$ is the complex modulus; $D^{\prime }$ is the storage compliance obtained from complex modulus test; $D^{\prime \prime }$ is the loss compliance obtained from complex modulus test; $lg\left|D^{\prime }\right|$ is the logarithm of the storage compliance; $\delta^{\prime \prime }$ is the value of the higher asymptote of the $lg\left|D^{\prime }\right|$ master curve; $\alpha^{\prime \prime }$ is the difference between the upper and lower asymptotes of the $lg\left|D^{\prime }\right|$ master curve; $\beta^{\prime \prime }$ and $\gamma^{\prime \prime }$ is shape coefficients of the $lg\left|D^{\prime }\right|$ master curve;. $\lambda^{\prime \prime }$ determined the model’s asymmetric characteristics (when $\lambda^{\prime \prime }\;$= 1 the shape of master curve is symmetric); $D^{\prime \prime }$ the master curve of loss compliance predicted by the GSM based on the approximate K-K relations; $k^{\prime \prime }$ is a positive correction, used to obtain a potentially more accurate prediction.

### 3.2. Determine the Discrete Relaxation Spectrum and Retardation Spectrum

The relaxation (retardation) time spectrum is the most general function describing the dependence of a material’s viscoelasticity on time or frequency [45]. It is the most vital part, and all viscoelastic functions can combine each other. Through studying the relaxation (retardation) time spectrum, we can obtain the distribution of the relaxation (retardation) time and the contribution of various motion modes to the macroscopic viscoelasticity [46], and providing an effective way to study the microstructure of viscoelastic materials.

#### 3.2.1. Determine the Discrete Relaxation Spectrum

The GMM [47] is widely used to characterize the relaxation behavior of asphalt mixtures in the LVE range. The relaxation modulus of the GMM expressed by the Prony series is shown in Equation (15).
(15)$$E\left(t\right)=E_{e}+\overset{N}{\underset{i=1}{\sum }}E_{i}e^{-\frac{t}{\rho_{i}}}=E_{g}-\overset{N}{\underset{i=1}{\sum }}E_{i}\left(1-e^{-\frac{t}{\rho_{i}}}\right)$$
where: E(t) is relaxation modulus, MPa.;$E_{e}$ is the equilibrium modulus, MPa; $E_{g}$ is the glass modulus, MPa; $E_{i}$ is the relaxation strength, MPa; $\rho_{i}$ is relaxation time, s; w is the angular frequency and is equal to 2πf.

The set of parameters consisting of all discrete relaxation times and corresponding relaxation strength is the discrete relaxation spectrum [33]. It is listed in Table A3 of Appendix B.

To obtain the discrete relaxation spectrum, the storage modulus and the loss modulus rewritten in the form of the Prony series, respectively. The error function $ef_{4}$ was applied to the test data of storage modulus and loss modulus. The equation was shown in Equation (16). The test data of storage modulus and loss modulus can be determined from the complex modulus test results following Equations (7) and (8). The Prony series parameters in Equations (17) and (18) for each mixture type was solved simultaneously by minimizing the error function.
(16)$$ef_{4}=ef_{E^{\prime }}+ef_{E^{\prime \prime }}=\frac{1}{N}\sqrt{\overset{N}{\underset{i=1}{\sum }}{\left(\frac{{\left|E^{\prime }\right|}_{ps,i}-{\left|E^{\prime }\right|}_{p,i}}{{\left|E^{\prime }\right|}_{p,i}}\right)}^{2}}+\frac{1}{N}\sqrt{\overset{N}{\underset{i=1}{\sum }}{\left(\frac{{\left|E^{\prime \prime }\right|}_{ps,i}-{\left|E^{\prime \prime }\right|}_{p,i}}{{\left|E^{\prime \prime }\right|}_{p,i}}\right)}^{2}}$$
where: $ef_{4}$ is the sum of error function of storage modulus and loss modulus; N is the number of the measured data points that is equal to 28; ${\left|E^{\prime }\right|}_{p,i}$ is the *i*-th data point predicted by storage modulus master curve of the GSM, MPa; ${\left|E^{\prime }\right|}_{ps,i}$ is the *i*-th data point of storage modulus predicted by Prony series, MPa; ${\left|E^{\prime \prime }\right|}_{p,i}$ is the *i*-th data point predicted by loss modulus master curve of the GSM, MPa; ${\left|E^{\prime \prime }\right|}_{ps,i}$ is *i*th data point of loss modulus predicted by Prony series.
(17)$$E^{\prime }\left(w\right)=E_{e}+\overset{m}{\underset{i=1}{\sum }}\frac{w^{2}\rho_{i}^{2}E_{i}^{2}}{w^{2}\rho_{i}^{2}+1}$$
(18)$$E^{\prime \prime }\left(w\right)=\overset{m}{\underset{i=1}{\sum }}\frac{w\rho_{i}E_{i}}{w^{2}\rho_{i}^{2}+1}$$
where: $E^{\prime }\left(w\right)$ and $E^{\prime \prime }\left(w\right)$ the storage modulus and the loss modulus rewritten in the form of the Prony series.

Before fitting the storage modulus and loss modulus with the Prony series, the GSM was used to smooth the storage modulus data, according to the generalized K-K relations, then get the master curve form of loss modulus. The expression has also been used to smooth the loss modulus data. Then the smoothed storage modulus and loss modulus data are respectively fitted for the Prony series. Finally, the discrete relaxation spectrum is obtained.

#### 3.2.2. Determine the Discrete Retardation Spectrum

The GKM [7] is widely used to characterize the creep behavior of asphalt mixtures in the LVE range. The creep compliance of the GKM expressed by the Prony series is shown in Equation (19).
(19)$$D\left(t\right)=D_{g}+\overset{n}{\underset{j=1}{\sum }}D_{j}\left(1-e^{-\frac{t}{\tau_{j}}}\right)=D_{e}-\overset{n}{\underset{j=1}{\sum }}D_{j}e^{-\frac{t}{\tau_{j}}}$$
where: D(t) is the creep compliance, MPa^−1^; $D_{g}$ is the equilibrium compliance, MPa^−1^; $D_{j}$ is retardation strength, MPa^−1^; $\tau_{j}$ is the retardation time, s is obtained by the Park and Schapery method [2]. The detailed selection process for discrete retardation time in the Prony series of mixtures is shown in Figure A1, Figure A2, Figure A3 and Figure A4 of Appendix B.

The set of parameters consisting of all discrete retardation times and retardation strengths is the discrete retardation spectrum. It is listed in Table A4 of Appendix B.

To obtain the discrete retardation spectrum, the storage compliance and the loss compliance rewritten in the form of the Prony series, respectively. The error function $ef_{5}$ was applied to the test data of storage compliance and loss compliance. The corresponding equation is shown in Equation (20). The test data of storage compliance and loss compliance can be determined from the complex modulus test results following Equation (12). The Prony series parameters in Equations (21) and (22) for each mixture type was solved simultaneously by minimizing the error function.
(20)$$ef_{5}=ef_{D^{\prime }}+ef_{D^{\prime \prime }}=\frac{1}{N}\sqrt{\overset{N}{\underset{i=1}{\sum }}{\left(\frac{{\left|D^{\prime }\right|}_{ps,i}-{\left|D^{\prime }\right|}_{p,i}}{{\left|D^{\prime }\right|}_{p,i}}\right)}^{2}}+\frac{1}{N}\sqrt{\overset{N}{\underset{i=1}{\sum }}{\left(\frac{{\left|D^{\prime \prime }\right|}_{ps,i}-{\left|D^{\prime \prime }\right|}_{p,i}}{{\left|D^{\prime \prime }\right|}_{p,i}}\right)}^{2}}$$
where: $ef_{5}$ is the sum of error function of storage compliance and loss compliance; N is the number of the measured data points that is equal to 28; ${\left|D^{\prime }\right|}_{p,i}$ is the *i*-th data point predicted by storage compliance master curve of GSM, MPa^−1^; ${\left|D^{\prime }\right|}_{ps,i}$ is the *i*-th data point of storage compliance predicted by Prony series, MPa^−1^; ${\left|D^{\prime \prime }\right|}_{p,i}$ is the *i*-th data point predicted by loss compliance master curve of GSM, MPa^−1^; ${\left|D^{\prime \prime }\right|}_{ps,i}$ is the *i*-th data point of loss compliance predicted by Prony series, MPa^−1^.
(21)$$D^{\prime }\left(w\right)=D_{g}+\overset{n}{\underset{j=1}{\sum }}\frac{D_{j}}{w^{2}\tau_{j}^{2}+1}$$
(22)$$D^{\prime \prime }\left(w\right)=\frac{1}{\eta_{0}w}+\overset{n}{\underset{j=1}{\sum }}\frac{w\tau_{j}D_{j}}{w^{2}\tau_{j}^{2}+1}$$
where: $D^{\prime }\left(w\right)$ and $D^{\prime \prime }\left(w\right)$ the storage compliance and the loss compliance rewritten in the form of the Prony series.

Similarly, before fitting the master curves of storage compliance and loss compliance with the Prony series, the GSM was used to smooth the storage compliance data; according to the generalized K-K relations, then get the expression form of the master curves of loss compliance based on GSM. The expression has also been used to smooth the loss compliance data. Finally, the smoothed storage compliance and loss compliance data are fitted in the Prony series, and then the discrete retardation spectrum is also obtained.

### 3.3. Determine the Continuous Relaxation Spectrum and Retardation Spectrum

Although the discrete-time spectrum can characterize the viscoelastic properties of asphalt mixtures, the discrete spectrum is dependent on the spacing of discrete-time. The viscoelastic response function obtained from the discrete-time spectrum is prone to oscillations in the local region, which could be solved by the continuous-time spectrum.

#### 3.3.1. Determine the Continuous Relaxation Spectrum

Integral transformation theory can be used to establish the relations between various LVE functions and the continuous-time spectrum. The continuous relaxation spectrum and the retardation spectrum can be derived from the corresponding storage modulus and storage compliance, respectively. The angular frequency w in storage modulus was replaced by $\rho^{-1}exp\left(\pm \frac{j\pi }{2}\right)$, and replaced by $\tau^{-1}exp\left(\pm \frac{j\pi }{2}\right)$ for storage compliance. Finally, only the imaginary part is retained.

Equation (23) is the expression of the continuous relaxation time spectrum. We can get it following the method proposed by Tschoegl [4].
(23)$$H\left(\rho \right)=\pm \left(\frac{2}{\pi }\right)ImE^{\prime }\left[\rho^{-1}\exp\left(\pm \frac{j\pi }{2}\right)\right]$$

Then we rewrite Equation (9) to form Equation (24).
(24)$$E^{\prime }\left(\omega \right)=\exp\left(A\right)\exp\left(\frac{B}{{\left[1+\lambda^{\prime }\exp\left(C+Dln\alpha_{T}+Dln\omega \right)\right]}^{\frac{1}{\lambda^{\prime }}}}\right)$$
where $A=\delta^{\prime }ln10;$
$A=\alpha^{\prime }ln10;$
$C=\beta^{\prime }-\gamma^{\prime }\log\left(2\pi \right);$
$D=\frac{\gamma^{\prime }}{ln10};$

Substituting Equation (24) into Equation (23), and according to Euler’s Equation, we obtain the function expression Equation (25) for the continuous relaxation spectrum.
(25)$$H\left(\rho \right)=\pm \left(\frac{2}{\pi }\right)\exp\left(A\right)Im\langle \exp\left\{{\left[P\left(\rho \right)\right]}^{\lambda^{\prime }}\right\}\rangle$$
where the expressions of P(ρ), X(ρ), and Y(ρ) are shown is Equations (26)–(28).
(26)P(ρ)=X(ρ)∓iY(ρ)
(27)$$X\left(\rho \right)=\frac{B^{\lambda^{\prime }}\left[1+\lambda^{\prime }\exp\left(E\right)cos\frac{\pi D}{2}\right]}{1+{\left(\lambda^{\prime }\right)}^{2}\exp\left(2E\right)+2\lambda^{\prime }\exp\left(E\right)cos\frac{\pi D}{2}}$$
(28)$$Y\left(\rho \right)=\frac{B^{\lambda^{\prime }}\lambda^{\prime }\exp\left(E\right)sin\frac{\pi D}{2}}{1+{\left(\lambda^{\prime }\right)}^{2}\exp\left(2E\right)+2\lambda^{\prime }\exp\left(E\right)cos\frac{\pi D}{2}}$$
where $E=C+D\ln\frac{\alpha_{T}}{\rho }$.

To further simplify Equation (25), when $F\left(\rho \right)={\left[X\left(\rho \right)\right]}^{2}+{\left[Y\left(\rho \right)\right]}^{2}$; $G\left(\rho \right)=arctan\frac{Y\left(\rho \right)}{X\left(\rho \right)}$; then Equation (25) is simplified to Equation (29).
(29)$$H\left(\rho \right)=-\left(\frac{2}{\pi }\right)\exp\left(A+F^{\frac{1}{2\lambda^{\prime }}}cos\frac{G}{\lambda^{\prime }}\right)\sin\left(F^{\frac{1}{2\lambda^{\prime }}}sin\frac{G}{\lambda^{\prime }}\right)$$

Finally, substituting the model parameters of the master curve of the storage modulus and the loss modulus into Equation (29), the function expression of the continuous relaxation time spectrum can be accurately obtained.

When *λ*′ = 1 Equation (29) can be further simplified to the form of Equation (30).
(30)$$H\left(\rho \right)=\left(\frac{2}{\pi }\right)\exp\left(A+X\left(\rho \right)\right)\sin\left(Y\left(\rho \right)\right)$$

#### 3.3.2. Determine the Continuous Retardation Spectrum

The continuous retardation spectrum can be derived from the master curve of storage compliance. The angular frequency *w* in the master curve of storage compliance was replaced by $\tau^{-1}\exp\left(\pm \frac{j\pi }{2}\right)$, and only the imaginary part was retained. Finally, the continuous retardation time spectrum was expressed as Equation (31).
(31)$$L\left(\tau \right)=\mp \left(\frac{2}{\pi }\right)ImD^{\prime }\left[\tau^{-1}\exp\left(\pm \frac{j\pi }{2}\right)\right]$$

Similarly, Equation (13) can be rewritten into the expression of Equation (32).
(32)$$D^{\prime }\left(\omega \right)=\exp\left(A^{\prime }\right)\exp\left(-\frac{B^{\prime }}{{\left[1+\lambda^{\prime \prime }\exp\left(C^{\prime }+D^{\prime }ln\alpha_{T}+D^{\prime }ln\omega \right)\right]}^{\frac{1}{\lambda^{\prime \prime }}}}\right)$$
where $A^{\prime }=\delta^{\prime \prime }ln10;$
$B^{\prime }=\alpha^{\prime \prime }ln10;$
$C^{\prime }=\beta^{\prime \prime }+\gamma^{\prime \prime }\log\left(2\pi \right);$
$D^{\prime }=-\frac{\gamma^{\prime \prime }}{ln10}$.

Then substituting Equation (32) into Equation (31), and according to Euler’s Equation, we obtain the function expression Equation (33) for the continuous retardation spectrum.
(33)$$L\left(\tau \right)=\mp \left(\frac{2}{\pi }\right)\exp\left(A^{\prime }\right)Im\langle \exp\left\{{\left[P\left(\tau \right)\right]}^{\lambda^{\prime \prime }}\right\}\rangle$$
where The expressions of P(τ), X(τ), and Y(τ) are shown in Equations (34)–(36).
(34)P(τ)=−X(τ)±iY(τ)
(35)$$X\left(\tau \right)=\frac{{B^{\prime }}^{\lambda^{\prime \prime }}\left[1+\lambda^{\prime \prime }\exp\left(E^{\prime }\right)sin\frac{\pi D^{\prime }}{2}\right]}{1+{\lambda^{\prime \prime }}^{2}\exp\left(2E^{\prime }\right)+2\lambda^{\prime \prime }\exp\left(E^{\prime }\right)cos\frac{\pi D^{\prime }}{2}}$$
(36)$$Y\left(\tau \right)=\frac{{B^{\prime }}^{\lambda^{\prime \prime }}\lambda^{\prime \prime }\exp\left(E^{\prime }\right)sin\frac{\pi D^{\prime }}{2}}{1+{\lambda^{\prime \prime }}^{2}\exp\left(2E^{\prime }\right)+2\lambda^{\prime \prime }\exp\left(E^{\prime }\right)cos\frac{\pi D^{\prime }}{2}}$$
where $E^{\prime }=C^{\prime }+D^{\prime }\ln\frac{\alpha_{T}}{\tau }$.

To simplify Equation (33), when $F\left(\tau \right)={\left[X\left(\tau \right)\right]}^{2}+{\left[Y\left(\tau \right)\right]}^{2}$; $G\left(\tau \right)=arctan\frac{Y\left(\tau \right)}{X\left(\tau \right)}$; then Equation (33) was simplified to Equation (37).
(37)$$L\left(\tau \right)=\left(\frac{2}{\pi }\right)\exp\left(A^{\prime }+F^{\frac{1}{2\lambda^{\prime \prime }}}cos\frac{G}{\lambda^{\prime \prime }}\right)\sin\left(F^{\frac{1}{2\lambda^{\prime \prime }}}sin\frac{G}{\lambda^{\prime \prime }}\right)$$

Finally, substituting the model parameters of the master curve of the storage compliance and the loss compliance into Equation (37), the function expression of the continuous retardation time spectrum can be accurately obtained.

When λ’’ = 1 Equation (37) can be further simplified to the form of Equation (38).
(38)$$H\left(\tau \right)=\left(\frac{2}{\pi }\right)\exp\left(A^{\prime }-X\left(\tau \right)\right)\sin\left(Y\left(\tau \right)\right)$$

## 4. Results and Discussion

### 4.1. Characterization of LVE Behavior of Crumb Rubber-modified Asphalt Mixtures

Solving Equation (3) can get the GSM fitting parameters of dynamic modulus and phase angle master curve, and also the parameter results were shown in Table 4. Figure 2 shows the master curves of the dynamic modulus and phase angle for the types of mixtures.

Solving Equation (6) can get the GSM fitting parameters for storage modulus and loss modulus, and also the parameter results are showed in Table 5. Figure 3 shows the master curves of the storage modulus and loss modulus for the types of mixtures.

Solving Equation (11) we can get the generalized model parameters of storage compliance and loss compliance master curve, and the corresponding parameter results are shown in Table 6. Figure 4 shows the master curves of the storage compliance and loss compliance for the types of mixtures.

### 4.2. Relaxation Spectrum and Retardation Spectrum of Asphalt Mixture

The relaxation (retardation) time spectrum is a useful tool for describing the viscoelasticity of asphalt and mixtures [48]. It is also the most general functional relations of viscoelasticity to time or frequency dependence. From the Boltzmann superposition principle, it is known that the full properties of the material are represented by all modes of motion that vary with the time spectrum. Moreover, the material functions measured in various experiments are based on the same relaxation (retardation) time spectrum, which is the vital framework of all viscoelastic materials. Many research results [49,50] show that the relaxation (retardation) spectrum is one of the essential characteristics of LVE materials, from which all LVE functions in the time and frequency domains can be derived.

#### 4.2.1. The Discrete Relaxation Spectrum and Retardation Spectrum of Asphalt Mixture

Figure 5 shows the discrete relaxation and retardation spectrum of asphalt mixture obtained using Equations (16)–(22). The bell-shaped relaxation and retardation spectrum peak intensities correspond to relaxation and retardation times located at around 10^−3^ s and 10^2^–10^3^ s, respectively. The relaxation spectrum peak point is consistent Gundla’s work [47]. When relaxation times more than 10^−3^ s, the discrete relaxation spectrum of HMA-60 is always below HMA-C. That is to say, the discrete relaxation spectrum strength of the former is always smaller than the latter when relaxation times more than 10^−3^ s. When the relaxation time is less than 10^−3^ s, the data for former and latter are too few to accurately compare, it is due to the limited of discrete time points. The discrete retardation spectrum of HMA-60 is always above the HMA-C, and the former has a broader retardation spectrum width than the latter. That is to say, the discrete retardation spectrum strength of the former is always higher than the latter. Considering the discrete relaxation spectra and discrete retardation spectra of the two mixtures (HMA-60 and HMA-C), it can be learned that the elastic component of HMA-60 is smaller than HMA-C and the viscous component is larger than HMA-C, which also indicates that HMA-C only needs a short time to achieve the retardation process under loading. This is because the viscosity of the compound-mesh rubber-modified asphalt binder is greater than the 60-mesh, then the cohesion of HMA-C is also greater than HMA-60, which gives HMA-C greater strength and deformation recovery. The discrete relaxation spectrum of the WMA-60 is always below the WMA-C. That is to say, the discrete relaxation spectrum strength of the former is always smaller than the latter. While the discrete retardation spectrum of WMA-60 is always above the WMA-C, and the former has a broader retardation spectrum width than the latter.. That is to say, the discrete retardation spectrum strength of the former is always higher than the latter. Considering the discrete relaxation spectra and discrete retardation spectra of the two mixtures (WMA-60 and WMA-C), it can be learned that the elastic component of WMA-60 is smaller than WMA-C and the viscous component is larger than WMA-C. Which also indicates that WMA-C only needs a short time to achieve the retardation process under loading. This is also because the viscosity of the compound-mesh rubber-modified asphalt binder is greater than the 60-mesh, then the cohesion of WMA-C is also greater than WMA-60, which gives WMA-C greater strength and deformation recovery. Compared the HMCRMA and WMCRMA, once warm mix additive was added, the intensity of the peak relaxation spectrum decreased, the width of the relaxation spectrum decreased, and the discrete relaxation spectrum was difficult to distinguish on the left side of the peak intensity. This means that the elastic component of the mixture is reduced and the viscous component increased, so the mixture needs a shorter time to achieve the relaxation process under load, which is due to the reduced aging effect after the warm mix additive was added.

#### 4.2.2. The Continuous Relaxation Spectrum and Retardation Spectrum of Asphalt Mixture

The continuous relaxation and retardation spectrum function of asphalt mixture are shown in Figure 6. The continuous relaxation and retardation spectrum also present a typical bell-shape in a the wide time domain. The continuous relaxation spectrum is asymmetric, further illustrating the advantages of using the GSM and approximate K-K relations to fit the master curve of the viscoelastic response function. The magnitude of continuous relaxation spectrum of HMA-60 is less than HMA-C when the reduced time is greater than 10^−3^ s and greater than HMA-C once the reduced time is less than 10^−3^ s, in other words, the continuous relaxation spectrum magnitude of the former is less than that of the latter in longer reduced time and greater than the latter in shorter reduced time. However, the magnitude of the former’s continuous retardation spectrum is always greater than the latter in all time domains. Furthermore, the former has a wider retardation spectrum width than the latter. In other words, the strength of the former’s continuous retardation spectrum is always greater than the latter in all time domains. Considering the continuous relaxation spectra and continuous retardation spectra of the two mixtures (HMA-60 and HMA-C), it can be learned that the elastic component of HMA-60 is smaller than HMA-C and the viscous component is larger than HMA-C, which also indicates that HMA-C only needs a short time to achieve the retardation process under loading. This is because the viscosity of the compound-mesh rubber-modified asphalt binder is greater than the 60-mesh, then the cohesion of HMA-C is also greater than HMA-60, which gives HMA-C greater strength and deformation recovery. The magnitude of the continuous relaxation spectrum of the WMA-60 is always less than the WMA-C. In other words, the continuous relaxation spectrum strength of the former is always smaller than that of the latter, while the magnitude of the continuous retardation spectrum of the former is always greater than the latter. That is to say, the strength of the continuous Retardation spectrum of the former is always greater than the latter. Considering the continuous relaxation spectra and continuous retardation spectra of the two mixtures (WMA-60 and WMA-C), it can be learned that the elastic component of WMA-60 is smaller than WMA-C and the viscous component is larger than WMA-C, which also indicates that WMA-C only needs a short time to achieve the retardation process under loading. This is because the viscosity of the compound-mesh rubber-modified asphalt binder is greater than the 60-mesh, then the cohesion of WMA-C is also greater than WMA-60, which gives WMA-C greater strength and deformation recovery. Comparing the hot-mixed and warm-mixed crumb rubber-modified asphalt mixtures, once warm mix additive was added, the width of the relaxation (retardation) spectrum narrowed, the peak intensity decreased, and The continuous retardation spectrum peak point shifted horizontally to the left. It indicates that the mixture’s elastic component is reduced and increased viscous component, so the mixture only needs a shorter time to achieve the relaxation process under load. The faster relaxation mechanism is due to the reduced aging effect after the warm mix additive was added [51]. From the molecular point, the primary reason for the above results can be attributed to the decrease in molecular weight and the concentration of polar functional groups in the asphalt binders [52].

#### 4.2.3. Compared the Discrete and Continuous Relaxation Spectrum and Retardation Spectrum of Asphalt Mixture

The inset in Figure 7 shows the discrete relaxation time where the Prony series have been plotted against the relaxation times. The continuous relaxation spectrum obtained from Equation (29) is also shown on the same plot as the dashed line. A similar plot is shown in the inset of Figure 8 for the retardation time.From the results in Figure 7 and Figure 8, we can see that the discrete spectrum of the crumb rubber-modified asphalt mixtures coincides with the continuous spectrum, Comparing the discrete, continuous relaxation and retardation spectrum of asphalt mixtures, the shape of the discrete-time spectrum is similar to that of the continuous-time spectrum. However, the peak point of the discrete relaxation spectrum is closer to the longer relaxation time than the continuous relaxation spectrum. conversely, the peak point of the discrete retardation spectrum is closer to the shorter retardation time than the continuous retardation spectrum. The most significant difference is that the discrete spectrum’s strength is more than the continuous spectrum at the same time. These results are consistent with the results of many other scholars [22,53].

In order to compare the discrete and continuous spectra more significantly, the continuous spectrum was converted to its corresponding discrete spectrum by conversion method, and then illustrated it with the discrete spectrum obtained from Prony series in Figure 9.

The discrete relaxation spectra can be efficiently calculated from continuous relaxation spectra following Equation (39).
(39)$$E_{i}^{\prime }=H\left(\rho_{i}\right)\Delta ln\rho_{i}$$
where $E_{i}^{\prime }$ is discrete relaxation spectrum strength transformed from continuous spectrum; $H\left(\rho_{i}\right)$ is continuous spectrum; $\Delta ln\rho_{i}$ is logarithmic interval of discrete relaxation time.

Similarly, the discrete retardation spectra can be efficiently calculated from continuous retardation spectra following Equation (40).
(40)$$D_{j}^{\prime }=L\left(\tau_{j}\right)\Delta ln\tau_{j}$$
where $D_{j}^{\prime }$ is discrete retardation spectrum strength transformed from continuous spectrum; $L\left(\tau_{j}\right)$ is continuous retardation spectrum; $\Delta ln\rho_{i}$ is logarithmic interval of discrete retardation time.

Although the shape of the discrete spectrum converted from the continuous spectrum is similar the discrete spectrum calculated from the Prony series, there are some differences in the local region, the reasons for this are as follows.

(1)The complex modulus test carried out in temperature of 5 °C–50 °C, and there are some errors in the test. The higher the temperature, the greater the error.(2)We used the approximate K-K relations rather than the exact K-K relations. In addition, the GSM constructed from the test results tend to show larger errors at extremely low and high frequencies.(3)In the process of using the Prony series to calculate the discrete spectrum, the discrete time is pre-selected, and the discrete spectrum intensity is obtained by the optimization algorithm, but it is a multi-solution problem, although we set constraints during the optimization process.

### 4.3. Construction of Master Curves Relaxation Modulus and Creep Compliance

Creep and stress relaxation are the essential phenomena of viscoelastic materials such as asphalt mixtures [54]. Creep is the phenomenon in which when a constant stress is applied, and the strain gradually increases with time. Stress relaxation is the phenomenon in which constant strain is applied, and the stress gradually decreases with time. Creep compliance and relaxation modulus are essential indicators for evaluating creep and relaxation. In the process of asphalt pavement design, creep compliance is an essential parameter for predicting strain, and relaxation modulus is an essential parameter for predicting stress [55]. However, the relaxation test has higher requirements on the instruments, and the strain is challenging to remain constant; besides, the creep tests typically require longer time to accurately obtain creep compliance. To solve this problem, the storage modulus and loss modulus master curves are obtained from the test results of the dynamic modulus test, then the discrete spectrum and continuous spectrum are obtained respectively according to the Prony series and Laplace inversion. Furthermore, the creep compliance and relaxation modulus obtained from the known discrete and continuous spectrum.

#### 4.3.1. The Relaxation Modulus and Creep Compliance Obtained from the Discrete Spectrum

According to Equation (15), the relaxation modulus can be obtained from the result of the discrete relaxation spectrum. similarly, according to Equation (19), the creep compliance can be obtained from the result of the discrete relaxation spectrum. All of the results are plotted in Figure 10.

Figure 10 shows that the relaxation modulus and creep compliance obtained from the discrete spectrum. It was Z-shaped in a broad time domain, which shows that the relaxation modulus tends to the maximum magnitude in the shorter time domain and the minimum magnitude in the longer time domain; while the creep compliance tends to the minimum magnitude in the shorter time domain and the maximum magnitude in the longer time domain. The results are consistent with many previous works [24,29].

Figure 10 also shown that the relaxation modulus and creep compliance obtained from the discrete-time spectrum change significantly just in the time domain of 10^−5^ s to 10^5^ s, but no significant change outside the region. Compared the relaxation modulus and creep compliance of hot mix crumb rubber-modified asphalt mixture, it found that the relaxation modulus of HMA-60 is always less than HMA-C in the entire time domain, which shows the stress relaxation ability of HMA-60 is better than HMA-C. The creep compliance of HMA-60 is higher than HMA-C in the entire time domain, which shows that the deformability of HMA-60 is better than HMA-C. This is because the viscosity of the 60-mesh rubber-modified asphalt binder is less than compound-mesh, then the cohesion of HMA-60 is also less than that of HMA-C, which gives HMA-60 a better relaxation and deformation capabilities. Compared the relaxation modulus and creep compliance of warm-mixed crumb rubber-modified asphalt mixture, it found that the relaxation modulus of WMA-60 is always less than that of WMA-C in the entire time domain, which shows the stress relaxation ability of WMA-60 is better than WMA-C. The creep compliance of WMA-60 is always higher than the WMA-C in the entire time domain, which shows that the deformability of WMA-60 is better than WMA-C. This is also because the viscosity of the 60-mesh rubber-modified asphalt binder is less than compound-mesh, then the cohesion of WMA-60 is also less than that of WMA-C, which gives WMA-60 a better relaxation and deformation capabilities. Comparing the hot-mixed and warm-mixed crumb rubber-modified asphalt mixtures, it found that once the warm mix additive was added, the relaxation modulus of crumb rubber-modified asphalt mixtures becomes smaller in the shorter time domain and larger in the longer time domain, creep compliance varies in the opposite way to the relaxation modulus. It shows that the addition of warm mix improves the asphalt mixture’s low-temperature stress relaxation ability and high-temperature deformation resistance. This is because the reduced aging effect after the warm mix additive was added. Finally, WMA-60 is recommended in the cold region, while WMA-C recommended in the hot region.

#### 4.3.2. The Relaxation Modulus and Creep Compliance Obtained from the Continuous Spectrum

The continuous relaxation spectrum can obtain the relaxation modulus, and the expression is as follows (Equation (41)).
(41)$$E\left(t\right)=E_{\infty }+\int_{-\infty }^{+\infty }H\left(\rho \right)\exp\left(-\frac{t}{\rho }\right)d\left(ln\rho \right)$$
where $E_{\infty }$ is the long terms equilibrium modulus, it can be determined by the method proposed by Liu [23], MPa; ρ is the relaxation time, s; *t* is the loading time, s.

The trapezoidal rule was applied to calculate the relaxation modulus. The integration interval is selected as (−30, 30), when $lg\rho_{n}$ > 30 and $lg\rho_{n}\;$ < −30, H(ρ) is so small that it can be regarded as 0. The expression is as follows (Equation (42)).
(42)$$E\left(t\right)\approx E_{\infty }+\left(ln10\right)\overset{n=M}{\underset{n=1}{\sum }}\frac{H\left(\rho_{n-1}\right)\exp\left(-\frac{t}{\rho_{n-1}}\right)+H\left(\rho_{n}\right)\exp\left(-\frac{t}{\rho_{n}}\right)}{2}\Delta lg\rho_{n}$$
where *M* is the number of subintervals in the integration interval, the value of *M* was 6000; $\Delta lg\rho_{n}$ is the length of the subinterval interval. In this paper, the value of $\Delta lg\rho_{n}$ was selected to be 0.01.

Then Equation (42) can be replaced by Equation (43), which is expressed as follows.
(43)$$\begin{array}{ll} E\left(t\right)\approx E_{\infty } & +\left(ln10\right)\frac{0.01}{2}\{\left[H\left(10^{-30}\right)\exp\left(-\frac{t}{10^{-30}}\right)+H\left(10^{-29.99}\right)\exp\left(-\frac{t}{10^{-29.99}}\right)\right]\\ & +\left[H\left(10^{-29.99}\right)\exp\left(-\frac{t}{10^{-29.99}}\right)+H\left(10^{-29.98}\right)\exp\left(-\frac{t}{10^{-29.98}}\right)\right]+\cdots \\ & +\left[H\left(10^{29.99}\right)\exp\left(-\frac{t}{10^{29.99}}\right)+H\left(10^{30}\right)\exp\left(-\frac{t}{10^{30}}\right)\right]\} \end{array}$$

Finally, the relaxation modulus can be obtained by the continuous relaxation time spectrum. The result was presented in Figure 11.

Similarly, according to Equation (44), the creep compliance can be obtained by continuous retardation spectrum.
(44)$$D\left(t\right)=D_{g}+\int_{-\infty }^{+\infty }L\left(\tau \right)\left[1-\exp\left(-\frac{t}{\tau }\right)\right]d\left(ln\tau \right)$$
where $D_{g}$ is the transient equilibrium compliance, it can be determined by Park and Schapery method, MPa; τ is the retardation time, s; *t* is the loading time, s.

The trapezoidal rule was applied to calculate creep compliance. The expression is as follows. The integration interval is selected as (−30, 30), when $lg\tau_{n}$ > 30 and $lg\tau_{n}\;$< −30, L(τ) is so small that it can be regarded as 0 (Equation (45)).
(45)$$D\left(t\right)\approx D_{g}+\left(ln10\right)\overset{n=M}{\underset{n=1}{\sum }}\frac{L\left(\tau_{n-1}\right)\left[1-\exp\left(-\frac{t}{\tau_{n-1}}\right)\right]+L\left(\tau_{n}\right)[1-\exp\left(-\frac{t}{\tau_{n}}\right)]}{2}\Delta lg\tau_{n}$$
where *M* is the number of subintervals in the integration interval, the value of *M* was 6000; $\Delta lg\tau_{n}$ is the length of the subinterval interval. in this paper, the value of $\Delta lg\tau_{n}$ was selected to be 0.01.

Then Equation (45) can be replaced by Equation (46), which is expressed as follows.
(46)$$\begin{array}{ll} D\left(t\right)\approx D_{g} & +(ln10)\frac{0.01}{2}\{L(10^{-30})[1-\exp\left(-\frac{t}{10^{-30}}\right)]\\ & +L(10^{-29.99})[1-\exp\left(-\frac{t}{10^{-29.99}}\right)]+L(10^{-29.99})[1\\ & -\exp\left(-\frac{t}{10^{-29.99}}\right)]+L(10^{-29.98})[1-\exp\left(-\frac{t}{10^{-29.98}}\right)]+\ldots \\ & +L(10^{29.99})[1-\exp\left(-\frac{t}{10^{29.99}}\right)]+L(10^{30})[1-\exp\left(-\frac{t}{10^{30}}\right)]\} \end{array}$$

Finally, Then the creep compliance can be obtained by the continuous retardation time spectrum. The result was also presented in Figure 11.

It can be learned from Figure 11 that the relaxation modulus and creep compliance obtained from the continuous time spectrum change significantly in the time domain of 10^−8^ s~10^7^ s, but do no change significantly outside that region. When the time domain is greater than 10^−5^ s, the relaxation modulus and creep compliance calculated by the continuous time spectrum are consistent with the discrete time spectrum. However, when the time domain is less than 10^−5^ s, the relaxation modulus and creep compliance calculated by the continuous time spectrum are different from the discrete time spectrum. The specific performance is that the relaxation modulus of HMA-60 in this region is greater than HMA-C, which shows that the stress relaxation ability of HMA-60 is less than HMA-C in the extremely shorter time domain (extremely low temperature). The creep compliance of HMA-60 is less than that of HMA-C in this region, which shows that the deformation ability of HMA-60 is not as good as HMA-C in low temperature. This is because the magnitude of the continuous relaxation spectrum of HMA-60 is significantly greater than HMA-C in shorter reduced times (lower temperatures), and the higher magnitude of the relaxation spectrum, the higher relaxation modulus.

#### 4.3.3. Compared Relaxation Modulus and Creep Compliance Obtained the Discrete and Continuous Spectrum

It can be learned from Figure 12 that the curve shape of the discrete spectrum modulus is similar to the continuous spectrum modulus. They have a Z-shape in the broad time domain. However, the discrete spectrum modulus changes significantly in the time domain of 10^−5^ s~10^5^ s, while the continuous spectrum modulus changes significantly in the time domain of 10^−8^ s~10^7^ s. The discrete spectrum modulus less than the continuous spectrum modulus in the shorter time domain (<10^−5^ s), but greater than the continuous spectrum modulus in the longer time domain (>10^−5^ s), and there is almost no difference between the discrete spectrum modulus and the continuous spectrum modulus in the middle time region (10^−5^ s~10^5^ s). The reason can be learned from Figure 9, where the magnitude of the relaxation spectrum obtained from the continuous relaxation spectrum is greater than the magnitude of the discrete relaxation spectrum in the lower time domain and less than the discrete relaxation spectrum in the higher time domain.

It can also be obtained in Figure 12 that the curve shape of the discrete spectrum compliance is also similar to the continuous spectrum compliance. They have a Z-shape in the broad time domain. However, the discrete spectrum compliance changes significantly in the time domain of 10^−5^ s~10^5^ s, while the continuous spectrum compliance changed significantly in the time domain of 10^−8^ s~10^7^ s. The discrete spectrum compliance higher than the continuous spectrum compliance in the shorter time domain (<10^−5^ s), but less than the continuous spectrum compliance in the longer time domain (>10^−5^ s), and there is almost no difference between the discrete spectrum compliance and the continuous spectrum compliance in the middle time region (10^−5^ s~10^5^ s). The strength of the retardation spectrum obtained by the continuous retardation spectrum is smaller than the discrete retardation spectrum strength in the lower time domain and greater than the discrete retardation spectrum strength in the higher time domain. The discrete spectrum modulus (compliance) is particularly close to the continuous spectrum modulus (compliance) in the time domain of 10^−5^ s~10^3^ s.

To compare the relative errors of the relaxation modulus and creep compliance constructed by discrete and continuous spectrum methods, respectively. The relative errors of the relaxation modulus and creep compliance can be defined follow the Equations (47) and (48), The result is presented in Figure 12.

The relative error of relaxation modulus.
(47)$$RE_{E}=\frac{\left|E_{d}\left(t\right)-E_{c}\left(t\right)\right|}{E_{c}\left(t\right)}$$
where: $RE_{E}$ is the relative errors of the relaxation modulus; $E_{d}\left(t\right)$ is the relaxation modulus calculated from discrete relaxation spectrum; $E_{c}\left(t\right)$ is the relaxation modulus calculated from continuous relaxation spectrum.

The relative error of creep compliance.
(48)$$RE_{D}=\frac{\left|D_{d}\left(t\right)-D_{c}\left(t\right)\right|}{D_{c}\left(t\right)}$$
where: $RE_{D}$ is the relative errors of the creep compliance; $D_{d}\left(t\right)$ is the creep compliance calculated from discrete retardation spectrum; $D_{c}\left(t\right)$ is the creep compliance calculated from continuous retardation spectrum.

It can be learned from Figure 13 that within the range of 10^−5^ s~10^3^ s, both the relaxation modulus and creep compliance calculated by the discrete spectrum are almost indistinguishable from those calculated by the continuous spectrum (the maximum error is only 4%), and once exceeded the range, there will be a significant difference (the maximum error for relaxation modulus and creep flexibility is 39% and 55%, respectively). The reason can be learned from Figure 9.

## 5. Conclusions

In this paper, based on the complex modulus test, we use the discrete and continuous spectrum to construct relaxation modulus and creep compliance master curves. The two methods are consistent with the LVE theory and allow for possible asymmetries in the spectrum curve. The following conclusions can be drawn from this study:(1)According to the complex modulus test data of four types of mixtures, the master curve models of storage modulus and loss modulus are developed according to the approximate K-K relations. These master curve models share the same model parameters and allow for possible asymmetry. Similarly, the master curve model of storage compliance and loss compliance is also developed according to the approximate K-K relations.(2)The continuous and discrete relaxation spectrum can be obtained from the storage modulus master curves utilizing integral transformations and the collocation method, respectively. Similarly, the continuous and discrete retardation spectrum can be obtained from the storage compliance master curve.(3)The shape of the discrete spectrum is similar to that of the continuous spectrum. The most significant difference is the intensity of the discrete spectrum is higher than that of the continuous spectrum.(4)The relaxation modulus and creep compliance were calculated by the discrete and continuous spectrum, respectively. The difference between the discrete and continuous spectrum is negligible in the time domain of 10−5 s-103 s. However, there are significant differences outside the region.(5)HMA-60 has a better deformation capacity than HMA-C at a shorter reduced time and a worse deformation resistance than HMA-C at a longer reduced time. Moreover, the deformation capacity of WMA-60 is better than WMA-C at a shorter reduced time, and a worse deformation resistance than WMA-C is seen at a longer reduced time. Once the warm mix additive was added, the mixture’s deformation capacity was improved at a shorter reduced time. Moreover, improved the deformation resistance in longer reduced time.

## Figures and Tables

**Figure 1 materials-13-03723-f001:**
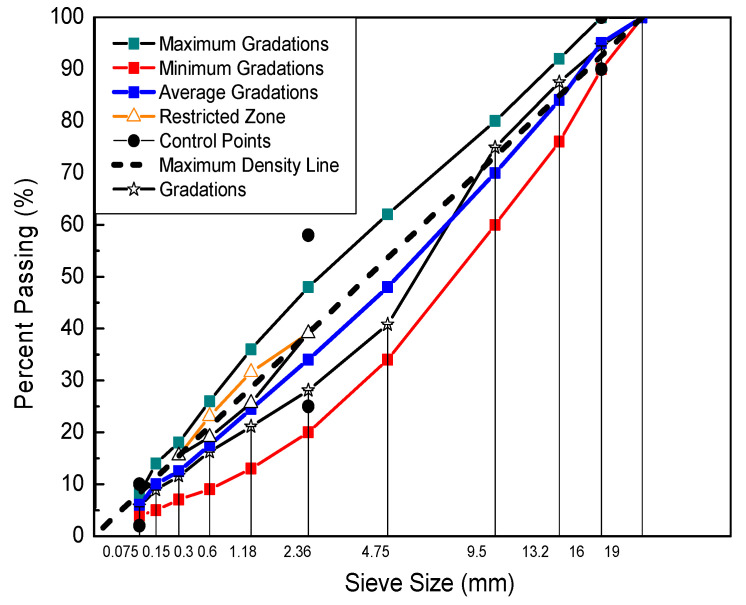
Crumb rubber-modified asphalt mixtures gradation.

**Figure 2 materials-13-03723-f002:**
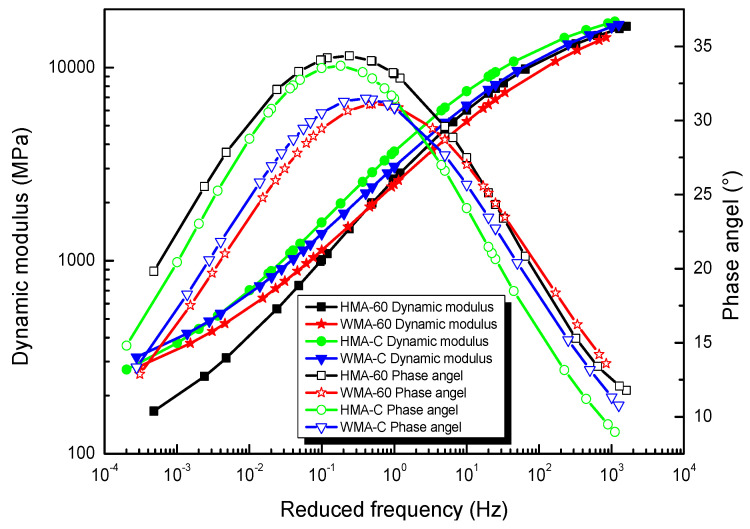
Master curve of dynamic modulus and phase angle.

**Figure 3 materials-13-03723-f003:**
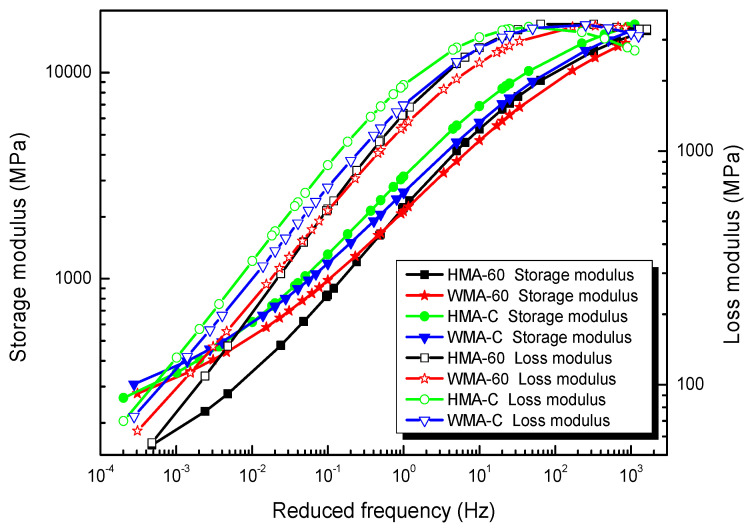
Master curve of storage modulus and loss modulus.

**Figure 4 materials-13-03723-f004:**
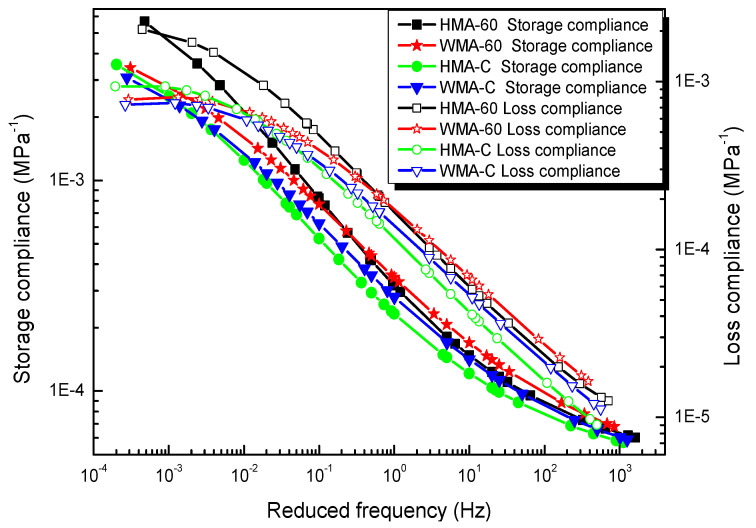
Master curve of storage compliance and loss compliance.

**Figure 5 materials-13-03723-f005:**
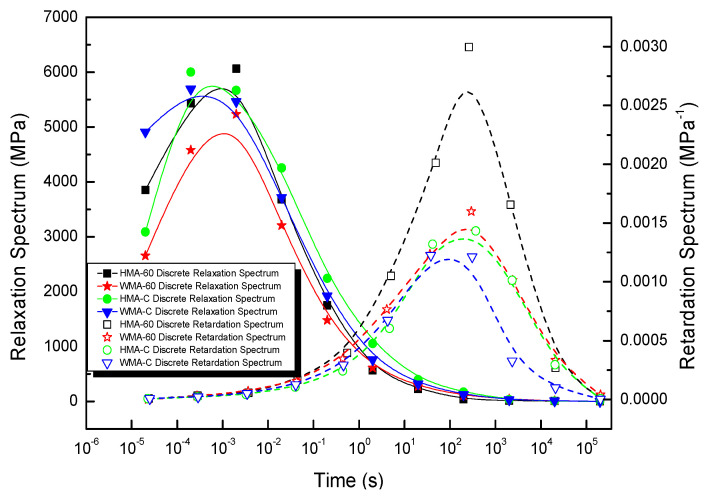
The discrete relaxation spectrum and retardation spectrum of asphalt mixture.

**Figure 6 materials-13-03723-f006:**
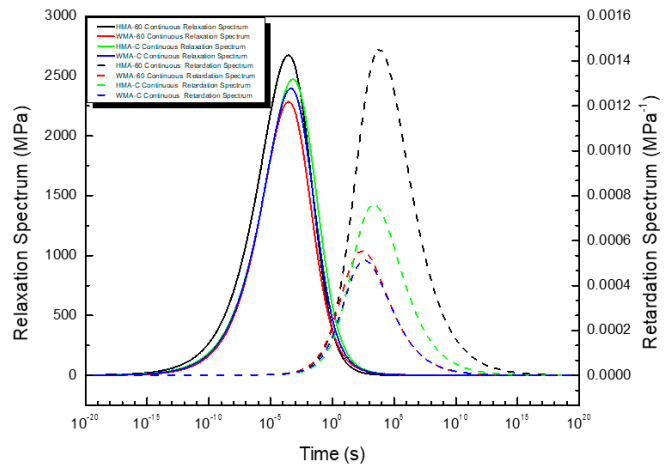
The Continuous relaxation spectrum and retardation spectrum of asphalt mixture.

**Figure 7 materials-13-03723-f007:**
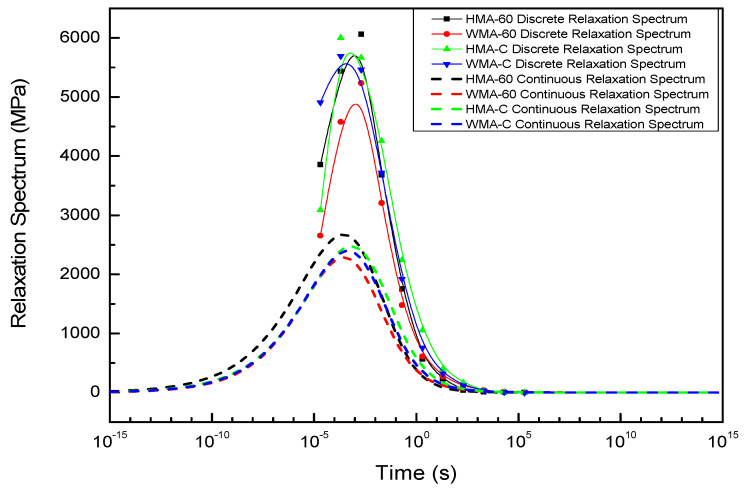
The discrete and continuous relaxation spectrum of asphalt mixture.

**Figure 8 materials-13-03723-f008:**
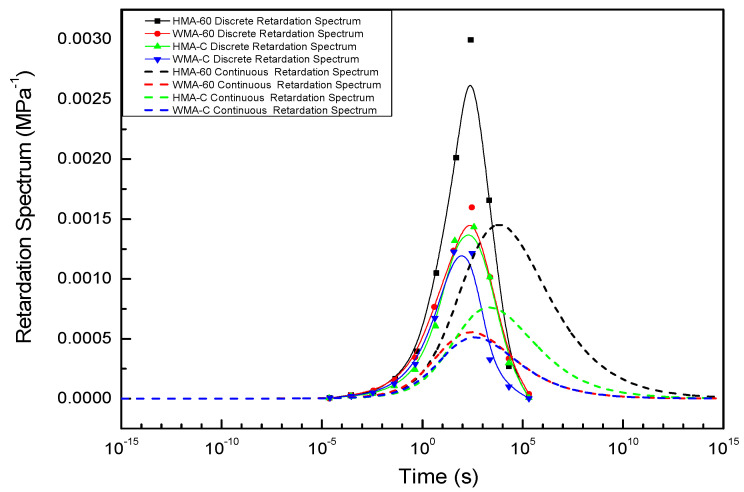
The discrete and continuous retardation spectrum of asphalt mixture.

**Figure 9 materials-13-03723-f009:**
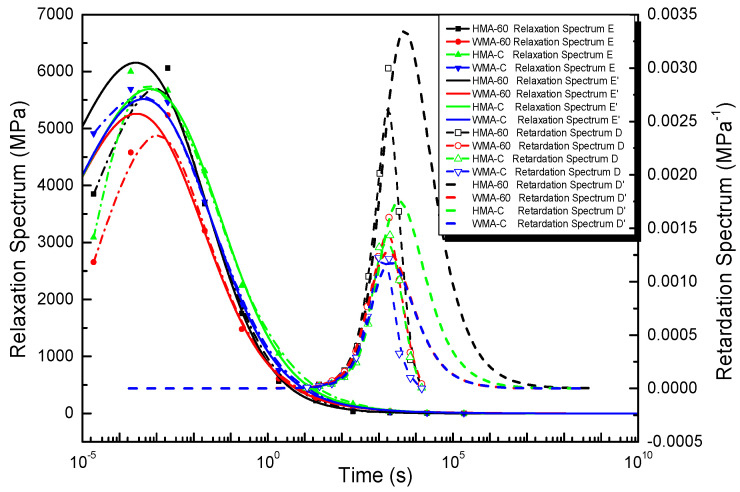
Compared the strength of discrete spectrum obtained from continuous spectrum and Prony series.

**Figure 10 materials-13-03723-f010:**
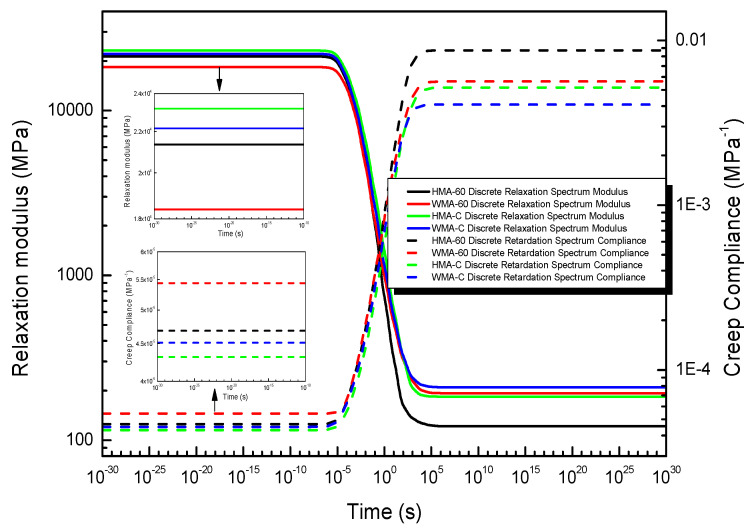
The relaxation modulus and creep compliance obtained from the discrete spectrum.

**Figure 11 materials-13-03723-f011:**
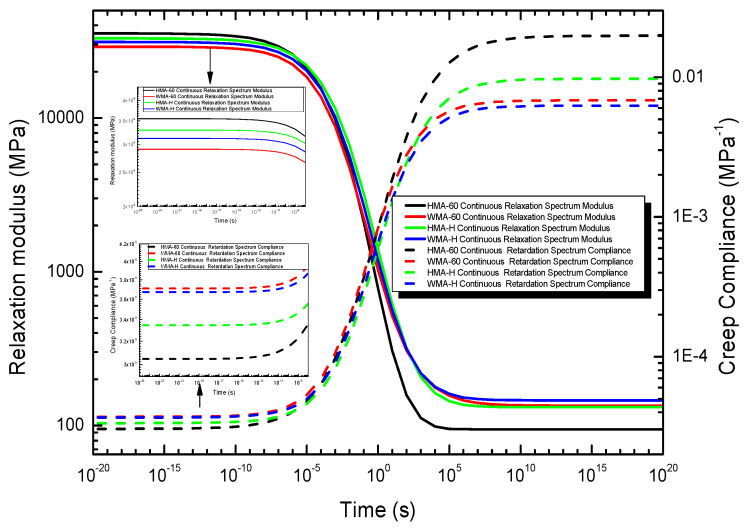
The relaxation modulus and creep compliance obtained from the continuous spectrum.

**Figure 12 materials-13-03723-f012:**
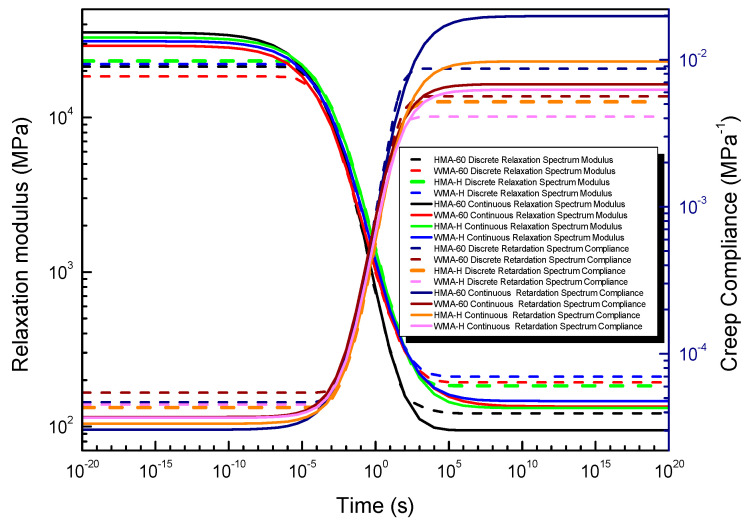
Compared the relaxation modulus and creep compliance obtained from the discrete and continuous spectrum.

**Figure 13 materials-13-03723-f013:**
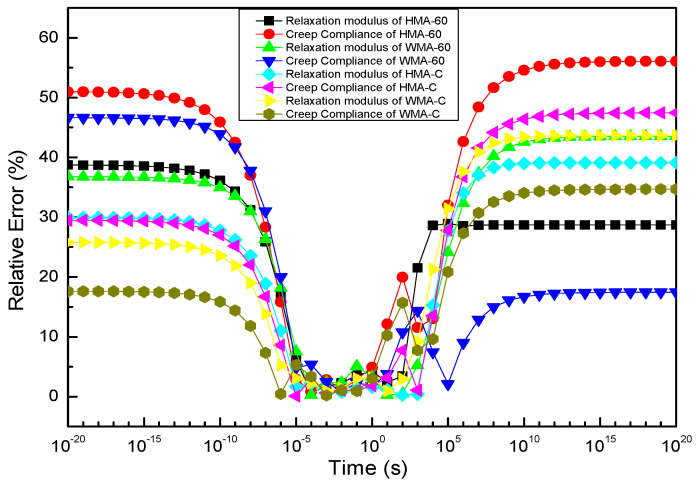
Relative errors of the relaxation modulus and creep compliance produced by methods of the discrete and continuous spectrum.

**Table 1 materials-13-03723-t001:** Aggregate stockpile blend percentages.

Aggregate	10–20 mm	5–10 mm	3–5 mm	0–3 mm	Filler
Blend Percentage by Weight/%	21	38	10	28	3

**Table 2 materials-13-03723-t002:** The softening points of the top and bottom of the samples after storing at 163 °C for 48 h.

Sample	Softening Point/°C		
	Bottom	Top	Difference
H-CR-60	63.5	57.0	6.5
W-CR-60	60.8	56.2	4.6
H-CR-C	68.1	59.8	8.3
W-CR-C	62.9	57.5	5.4

**Table 3 materials-13-03723-t003:** Volume parameters of optimum asphalt content.

Asphalt Mixture	OAC/%	Gross Density g/cm^3^	Theoretical Density g/cm^3^	Void/%	VMA/%	VFA/%	Stability/KN	Flow Value/mm
HMA-60	5.4	2.439	2.537	3.86	14.04	72.5	10.45	2.74
WMA-60	5.4	2.442	2.538	3.78	13.93	72.8	10.95	2.87
HMA-C	5.6	2.441	2.546	4.12	14.15	70.9	11.23	2.45
WMA-C	5.6	2.444	2.546	4.01	14.04	71.5	11.61	2.62
Standard	-	-	-	3~5	≥13	65~75	≥8	2~4

**Table 4 materials-13-03723-t004:** GSM fitting parameters for dynamic modulus and phase angle.

Mixture Type	Parameters								Error/%
	*δ*	*α*	*β*	*γ*	*λ*	*C* _1_	*C* _2_	*k*	
HMA-60	1.62	2.84	−0.60	−0.56	0.80	8.93	87.75	0.90	1.23
WMA-60	2.29	2.13	−0.14	−0.62	0.81	19.14	202.14	0.97	0.83
HMA-C	2.12	2.40	−0.54	−0.51	0.51	19.13	187.95	0.98	0.81
WMA-C	2.33	2.14	−0.31	−0.56	0.52	14.02	138.11	1.01	0.75

**Table 5 materials-13-03723-t005:** GSM fitting parameters for storage modulus and loss modulus.

Mixture Type	Parameters								Error/%
	*δ*′	*α′*	*β′*	*γ′*	*λ′*	*C* _1_	*C* _2_	*k′*	
HMA-60	1.98	2.57	−0.36	−0.51	0.22	8.93	87.75	1.00	1.12
WMA-60	2.13	2.33	−0.12	−0.58	0.92	19.14	202.14	1.00	0.83
HMA-C	1.86	2.66	−0.54	−0.53	0.79	19.13	187.95	1.00	0.81
WMA-C	2.16	2.33	−0.28	−0.55	0.67	14.02	138.11	1.00	0.82

**Table 6 materials-13-03723-t006:** GSM fitting parameters for storage compliance and loss compliance.

Mixture Type	Parameters								Error/%
	*δ*″	*α*″	*β*″	*γ*″	*λ*″	*C* _1_	*C* _2_	*k*″	
HMA-60	−1.70	2.81	−0.55	0.53	1.00	8.93	87.75	1.15	1.21
WMA-60	−2.16	2.26	−0.27	0.60	1.00	19.14	202.14	1.06	0.72
HMA-C	−2.01	2.46	−0.61	0.57	1.00	19.13	187.95	1.08	0.90
WMA-C	−2.20	2.23	−0.42	0.61	1.00	14.02	138.11	1.16	0.72

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
