# Peer review of "The Discrete and Continuous Retardation and Relaxation Spectrum Method for Viscoelastic Characterization of Warm Mix Crumb Rubber-Modified Asphalt Mixtures"

_materials, 2020, doi:10.3390/ma13173723_

Round 1

Reviewer 1 Report

The article "Discrete and continuous delay and relaxation spectrum method for characterizing viscoelastic asphalt mixtures of a warm modified crumb mix" is an interesting collection of research and analyzes related to the application of the spectral method in asphalt technology. Scientific research completes the current state of knowledge. Nevertheless, after reading this article, some main considerations were defined:

  1. In line 116 the authors write:

The content of crumb rubber modified asphalt of warm mix asphalt mixture is the same with the corresponding hot mix asphalt mixture. HMA was mixed and compacted at 180 ℃ and 170 ℃, respectively. While WMA was mixed and compacted at 162 ℃ and 152 ℃, respectively.

The asphalt mix produced at 162 ℃ and 152 ℃ cannot be classified as WMA technology. The WMA technology is characterized by the production and incorporation of asphalt at a temperature of less than 140oC (Olard, F. and Noan, C., (2008). Low Energy Asphalts, Routes Roads, 336/337, PIARC, pp. 131-145). Therefore, authors should update the title of the article,

  1. Table 2 lacks the parameters of the WMA-60 and WMA-C mix,

  1. Lack of information on the statistical analysis of the test results of the types of asphalt mixture,

  1. The descriptions (legend) in Fig. 1-12 are illegible,

  1. Conclusions should be specifically formulated in points.

Author Response

Response to Reviewer 1 Comments

Point 1: In line 116 the authors write:

The content of crumb rubber modified asphalt of warm mix asphalt mixture is the same with the corresponding hot mix asphalt mixture. HMA was mixed and compacted at 180 ℃ and 170 ℃, respectively. While WMA was mixed and compacted at 162 ℃ and 152 ℃, respectively.

The asphalt mix produced at 162 ℃ and 152 ℃ cannot be classified as WMA technology. The WMA technology is characterized by the production and incorporation of asphalt at a temperature of less than 140oC (Olard, F. and Noan, C., (2008). Low Energy Asphalts, Routes Roads, 336/337, PIARC, pp. 131-145). Therefore, authors should update the title of the article,

Response 1: Thank you for your remarkable question. The mixed and compacted temperature of conventional mixtures often less than 175°C. But the hot mix crumb rubber-modified asphalt mixture is different from the conventional mixture, its mixing and compaction temperature are greater than the conventional mixture. although the warm mix technology was used, it still produced in higher temperatures than the conventional warm mix asphalt mixture. Despite the inconsistency with the literature (Olard, F. and Noan, C., (2008). Low Energy Asphalts, Routes Roads, 336/337, PIARC, pp. 131-145), but I don't know how to name it, so I have to follow the literature [[1]] to give a name.

Point 2: Table 2 lacks the parameters of the WMA-60 and WMA-C mix,

Response 2: Thank you for your remarkable question. The parameters of the WMA-60 and WMA-C mix have been supplemented according to your comments, the results were illustrated in Table 3 In line 192

Table 3 Volume parameters of optimum asphalt content

Asphalt

Mixture

OAC/%

Gross density g/cm³

Theoretical density g/cm³

Void/ %

VMA/%

VFA/%

Stability/KN

Flow value/

mm

HMA-60

5.4

2.439

2.537

3.86

14.04

72.5

10.45

2.74

WMA-60

5.4

2.442

2.538

3.78

13.93

72.8

10.95

2.87

HMA-C

5.6

2.441

2.546

4.12

14.15

70.9

11.23

2.45

WMA-C

5.6

2.444

2.546

4.01

14.04

71.5

11.61

2.62

Standard

3~5

≧13.

65~75

≧8

2~4

Point 3: Lack of information on the statistical analysis of the test results of the types of asphalt mixture,

Response 3: Thank you for your remarkable question. In accordance with your advice, we have added information on the statistical analysis of the test results of the four types of asphalt mixture. The results are presented in Table A1, A2. In line 1127-1183.

Coefficients of variation were computed to assess the variability of the test results of the types of asphalt mixture. Coefficient of variation CV was defined as follow:

Where  is coefficient of variation;  is standard deviation;  is the mean value;  is experimental results; n is number of experimental.

Coefficients of variation for measure dynamic modulus and phase angle at each temperature and frequency are listed in Table A1, A2.

Table A1 Coefficients Variation of Dynamic Modulus

Temperature

/

Coefficients Variation of Dynamic Modulus / %

0.1Hz

0.5Hz

1Hz

5Hz

10Hz

20Hz

25Hz

Average

HMA-60

5

3.5

3.9

4.6

4.9

5.7

5.7

6.5

4.97

20

5.1

5.6

6.4

8.3

8.5

9.5

8.7

7.44

35

10.4

11.3

11.5

11.6

12.4

14.2

15.9

12.47

50

12.8

12.8

13.9

14.6

15.4

16.9

16.8

14.74

WMA-60

5

4.1

4.6

5

5.2

4.9

5.6

6.7

5.16

20

4.8

5.3

6.7

7.9

7.8

9.5

10.6

7.51

35

9.9

10.6

13.4

11.1

14.8

14.5

17.2

13.07

50

11.4

12.9

14.6

14.8

15.3

16.2

17.3

14.64

HMA-C

5

3.2

4.9

6.2

5.5

7.1

6.9

10.4

6.31

20

5.6

5.1

6.7

7.9

9.3

10.9

12.4

8.27

35

9.9

11.3

10.7

10.4

11.6

10.9

13.7

11.21

50

11.9

13.5

12.4

11.7

12.8

12.7

14.5

12.79

WMA-C

5

4.5

5.5

4.4

5.1

5.7

6.3

8.7

5.74

20

6.3

5.2

7.1

7.5

9.2

8.5

10.8

7.80

35

9.9

11.3

10.9

10.4

11.7

13.8

15.1

11.87

50

11.5

11.8

12.9

13.7

12.4

11.8

15.9

12.86

Table A2 Coefficients Variation of Phase Angle

Temperature

/℃

Coefficients Variation of Phase Angle / %

0.1Hz

0.5Hz

1Hz

5Hz

10Hz

20Hz

25Hz

Average

HMA-60

5

5.6

6.2

7.1

7.8

8.2

9.2

10.3

7.77

20

11.5

12

13.4

14.2

15.3

16.7

17.5

14.37

35

16.1

17.6

17.2

20.3

21.7

20.9

22.3

19.44

50

18.5

21.6

20.4

19.8

22.3

21.5

20.8

20.70

WMA-60

5

6

6.8

7.3

7.5

8.3

8.9

9.9

7.81

20

10.4

11.6

12.8

13.9

15.7

15.9

18

14.04

35

15.3

16.7

17.3

19.9

19.4

20.3

21.1

18.57

50

17.6

20.5

20.6

18.9

19.8

22

21.5

20.13

HMA-C

5

6.3

6.8

7.0

8.3

8.8

9.1

10.6

8.13

20

10.7

11.7

12.9

13.5

14.9

15.7

16.5

13.70

35

15.4

16.8

17.2

20.5

21.2

20.6

21.8

19.07

50

17.9

21.4

20.3

18.9

21.7

20.5

22.1

20.40

WMA-C

5

4.5

5.5

6.1

6.9

8.4

8.6

9.7

7.10

20

10.3

11.2

12.1

13.5

14.2

15.5

16.8

13.37

35

15.4

16.9

18.2

17.6

20.4

20.3

21.6

18.63

50

16.9

20.7

22.7

20.8

21.5

20.9

21.5

20.71

The coefficients of variation for both dynamic modulus and phase angle increase

with increasing temperature, it is because the material properties are more difficult to measure as the asphalt binder becomes softer. The values for the present study listed in Table A1 and Table A2 are lower than Pellinen’s [[2]]. It is believed that the reasons for the lower variations in the present study are the high accuracy of the UTM testing machine and the less than desirable noise in the strain measurement signals. The tests were particularly difficult to control at 50℃; this is reflected in the large coefficients of variation for both dynamic modulus and phase angle.

Point 4: The descriptions (legend) in Fig. 1-12 are illegible,

Response 4: Thank you for your remarkable question. the legend of Figures 1-12 has been revised according your advice.

Point 5:  Conclusions should be specifically formulated in points.

Response 5: Thank you for your remarkable question. We revised the conclusions and made them specifically formulated in points as your advice. In line 1091-1114.

References

[1] Franesqui, M. A., Yepes, J., García-González, C., & Gallego, J. (2019). Sustainable low-temperature asphalt mixtures with marginal porous volcanic aggregates and crumb rubber modified bitumen. Journal of Cleaner Production, 207, 44-56.

[2] Pellinen, T.K., Investigation of the Use of Dynamic Modulus as Indicator of Hot-Mix Asphalt Performance, Arizona State University, 2001, Tempe, AZ

Reviewer 2 Report

GENERAL COMMENTS

The paper deals with a mathematical modelling of the viscoelastic properties of asphalt rubber mixtures prepared using a warm technology. The modelling is performed using both the discrete and the continuous retardation and relaxation spectrum.

Overall, the paper seems a pure mathematical exercise without relevant practical implications for researchers and applicants. Such theoretical procedure is also affected by some interpretation errors. Moreover, justification of the observed findings is often missing. For example, based on the limited information reported, it seems that the only difference between mix -60 and mix -C is 0.2% bitumen content. If it is the case, the two mixtures are almost identical and thus the performance differences measured and modeled in the following (see Figs. 2, 3, etc.) do not seem reasonable. (see also specific comments below)

The use of English must be significantly improved.

Even if the introduction is effective and refers to the relevant literature, the objective of the paper should be clarified and better specified. Is the objective related to the specific type of tested mixtures (warm, AR) or only focused on the comparison between the discrete and continuous spectra? Etc.

The description of the materials and methods must be significantly improved. In the current version, a colleague cannot reproduce the experiments (see also specific comments below). Improvements are recommended also regarding the presentation of the theoretical background.

Conclusions are affected by the abovementioned issues.

SPECIFIC COMMENTS

Line 67: reference 12 is not reported

Line 78: ? – Lines 93-94: ? – Lines 122-124: ? – Lines 414-427: ? – etc. please improve significantly the use of English

All the acronyms should be spelled out the first time they appear, included the well-known ones (e.g. AC, HMA, etc.)

What about type and properties of aggregates, warm additive and crumb rubber?

What about the preparation procedure of the asphalt rubber binder? (e.g. temperature, time, rotation speed, storage stability, etc.)

Section 2.2: have you checked if the specimens experienced some damage at the end of the frequency sweeps? Have you verified that 70 microstrain effectively assured the linear viscoelastic domain? Please specify

All the symbols introduced in the equations should be explained in the text immediately after the related equations. All the symbols

The second term of equations 3 and 8 is the same while the first is different. Please check. Similar discrepancies are also presented in some other equations. Please check all again and carefully. And also explain better how the different “sets” of equations are related each other.

Line 173: equation 5??

Figure 5: please use a blue curve for WMA-C, like the color of the indicator

Lines 362-378: not clear and not thoroughly analyzed

Lines 382-405: justification?

Figures 7 and 8: please use dashed/dotted lines for the continuous spectra

Figure 9 joins the data reported in Figures 7 and 8; thus you should use only Figure 9 OR both Figure 7 and 8

Lines 414-415: ?? the discrete spectra did not seem coincident with the discrete spectra

Lines 416-418: ??

Lines 420-422: ??

Lines 436-437: what does it mean “the creep test takes a longer time”?

Lines 459-472: justification?

Line 505: Figure 14 or 13?

Figure 12 also reports the data depicted in Figure 11; thus you should use only Figure 12

Lines 510-514: this is not showed in Figure 11

Lines 514-518: justification?

Lines 528-531: justification?

Lines 535-551: justification?

Figure 13: have you already defined the strength?

Figure 14: have you already defined the relative error?

Lines 573-574: ???

Author Response

Response to Reviewer 2 Comments

The paper deals with a mathematical modelling of the viscoelastic properties of asphalt rubber mixtures prepared using a warm technology. The modelling is performed using both the discrete and the continuous retardation and relaxation spectrum.

Point 1: Overall, the paper seems a pure mathematical exercise without relevant practical implications for researchers and applicants. Such theoretical procedure is also affected by some interpretation errors. Moreover, justification of the observed findings is often missing. For example, based on the limited information reported, it seems that the only difference between mix -60 and mix -C is 0.2% bitumen content. If it is the case, the two mixtures are almost identical and thus the performance differences measured and modeled in the following (see Figs. 2, 3, etc.) do not seem reasonable. (see also specific comments below)

Point 2: The use of English must be significantly improved.

Response 2: Thank you for your remarkable question. In accordance with your advice, we have checked and carefully revised the English language and grammar of the entire paper.

Point 3: Even if the introduction is effective and refers to the relevant literature, the objective of the paper should be clarified and better specified. Is the objective related to the specific type of tested mixtures (warm, AR) or only focused on the comparison between the discrete and continuous spectra? Etc.

Response 3: Thank you for your remarkable question. In accordance with your advice, we have supplemented the objective of the paper in the introduction. The objective of this paper is to comparing the relaxation modulus and creep compliance master curves of crumb rubber modified asphalt mixtures constructed by discrete and continuous spectrum methods and to evaluate the accuracy of the master curve model.

Point 4: The description of the materials and methods must be significantly improved. In the current version, a colleague cannot reproduce the experiments (see also specific comments below). Improvements are recommended also regarding the presentation of the theoretical background.

Response 4: Thank you for your remarkable question. In accordance with your advice, we have significantly improved the description of materials and methods. In addition, the presentation of the theoretical background has also been improved.

Point 5: Conclusions are affected by the abovementioned issues.

Response 5: Thank you for your remarkable question. We revised the conclusions and made them specifically formulated in points as your advice.

The revised conclusions are as follows:

In this paper, based on the complex modulus test, we use the discrete and continuous spectrum to construct relaxation modulus and creep compliance master curves. The two methods are consistent with the LVE theory and allow for possible asymmetries in the spectrum curve. The following conclusions can be drawn from this study.

  • According to the complex modulus test data of four types of mixtures, the master curve model of storage modulus and loss modulus is developed according to the approximate K-K relations. These master curve models shared the same model parameters and allowed for possible asymmetry. Similarly, the master curve model of storage compliance and loss compliance is also developed according to the approximate K-K relations.
  • The continuous and discrete relaxation spectrum can be obtained from the Storage modulus master curves utilizing integral transformations and the collocation method, respectively. Similarly, the continuous and discrete retardation spectrum can be obtained from the storage compliance master curve.
  • The shape of the discrete spectrum is similar to the continuous spectrum. The most significant difference is the intensity of the discrete spectrum is higher than of the continuous spectrum.
  • The relaxation modulus and creep compliance were calculated by the discrete and continuous spectrum, respectively. The difference between the discrete and continuous spectrum is negligible in the time domain of 10-5 s-103 s. However, significant differences outside the region.
  • HMA-60 has a better deformation capacity than HMA-C at shorter reduced time and a worse deformation resistance than HMA-C at longer reduced time. Moreover, the deformation capacity of WMA-60 is better than WMA-C at shorter reduced time, and a worse deformation resistance than WMA-C at longer reduced time. Once the warm mix additive was added, the mixture's deformation capacity was improved at a shorter reduced time. Moreover, improved the deformation resistance in longer reduced time.

SPECIFIC COMMENTS

Point 6: Line 67: reference 12 is not reported

Response 6: Thank you for your remarkable question. In accordance with your advice, we have checked all of the references and delete the not reported reference 12.

Point 7: Line 78: ? – Lines 93-94: ? – Lines 122-124: ? – Lines 414-427: ? – etc. please improve significantly the use of English

Response 7: Thank you for your remarkable question. In accordance with your advice, the use of English in lines 78, 93-94, 122-124, and 414-427, etc. has been greatly improved. The details are as follows

the sentence "It is because the discrete spectrum has many problems. The continuous spectrum has been gradually developed in recent years " in line 78 has been modified to "Considering many problems with discrete spectrum. The continuous spectrum has been gradually developed in recent years".

The sentence " Despite the unprecedented development of methods for calculating discrete and continuous spectrum, the greater attention paid to the interconversion of each other" in lines 93-94 has been modified to "Although the Computational methods of discrete and continuous spectrum has achieved an unprecedented development, more and more people pay much attention to the Interconversion of each other".

The sentence" The specimen air void content was controlled between 3.5% and 4.5%. Three replicate specimens were fabricated and tested for every mixture type. All specimens were kept in an unlit cabinet until the test to reducing aging but not allowed for more than two weeks." in lines 122-124 has been modified to " The air void content of all type mixture was 4±0.5%. Three replicate specimens were fabricated and tested for every mixture type. Prior to testing all specimens were stored in an unlit cabinet to reduce ageing, but not allowed to more than two weeks ".

The sentence" It indicates that the mixture's elastic component is reduced and increased viscous component, so the mixture only needs a shorter time to achieve the relaxation process under load. The faster relaxation mechanism is due to the reduced aging effect after the warming additives were added. From the molecular point, the primary reason for the above results can be attributed to the decrease in molecular weight and the concentration of polar functional groups in the asphalt binders. " in line 414-427 has been modified to " The air void content of all type mixture was 4±0.5%. Three replicate specimens were fabricated and tested for every mixture type. Prior to testing all specimens were stored in an unlit cabinet to reduce ageing, but not allowed to more than two weeks ".

Point 8: All the acronyms should be spelled out the first time they appear, included the well-known ones

Response 8: Thank you for your remarkable question. In accordance with your advice, we have checked and carefully revised this article and added the full name of all the acronyms the first time they appear.

For example: generalized Maxwell model (GMM); generalized Kelvin model (GKM); generalized Sigmoidal model (GSM); linear viscoelastic (LVE); The Time-Temperature Superposition Principle (TTSP), et al.

Point 9: What about type and properties of aggregates, warm additive and crumb rubber?

Response 9: Basalt for coarse aggregates, limestone for fine aggregates, all aggregates from a quarry located in Zhuozishan, Inner Mongolia Autonomous Region, China. The results were shown in table 1-3. The warm mix additive (SDYK) was a type of surfactant, the content is 1% by the weight of asphalt binder. The technical parameters of warm mix additive were shown in Table 4. The crumb rubber produced by mechanical shredding at ambient temperature was obtained from same source of bias tire. crumb rubber of 60-mesh and compound-mesh are used respectively, and the gradation and technical parameters are shown in Table 5, 6.

Table 1 Results of Coarse aggregate

Items

Unit

Results

10mm~20mm

5mm~10mm

method

Crushed stone value

%

11

13

T0316

Los Angeles abrasion value

%

8.7

10.5

T0317

apparent specific gravity

2.932

2.948

T0304

water absorption

%

0.65

0.86

T0304

ruggedness

%

2

2

T0314

clay content

%

0.06

0.15

T0310

Table 2 Test results of fine aggregate

Items

Unit

Results

Standard value

Method

apparent specific gravity

2.723

2.5

T0328

ruggedness(>0.3mm)

%

25

12

T0340

clay content

%

0.6

3

T0333

Sand equivalent value

%

87

60

T0334

Methylene blue value

g/kg

1.8

25

T0346

angularity

s

46.3

30

T0345

Table 3 Test results of fines

Items

Unit

Result

Standard value

Method

Apparent density

t/m³

2.698

2.5

T0352

Water content

%

0.6

1

T0103

Sieves size

<0.6

%

100

100

T0351

<0.15

%

99.8

90~100

<0.075

%

99.5

75~100

Appearance

agglomeration-free

agglomeration-free

hydrophilic coefficient

0.5

<1

T0353

plasticity index

%

2.1

<4

T0354

Heating Stability

No change

Measured

T0355

Table 4 Technical parameters of warm mix additive

Property

Experiment result

Standard value

Appearance @25 ℃

Yellow liquid

Yellow liquid

Viscosity / mPa.s @25 ℃

650

500~1000

PH

12.0

11.5±1

Amine / (mg KOH/g)

550

510~610

Surface Tension (0.6g/L) / (mN/m) @25 ℃

29

≤40

Table 5 The gradation of crumb rubber

Sieve No.(um)

60 mesh crumb rubber

compounded mesh crumb rubber

Retained (%)

Cumulative retained (%)

Retained (%)

Cumulative retained (%)

40 (425)

0

0

37.8

37.8

60 (250)

9.2

9.2

40.1

77.9

80 (180)

32.5

41.7

13.0

90.9

100 (150)

33.7

75.4

7.3

98.2

120 (125)

10.6

86.0

1.3

99.5

>120 (>75)

14

100

0.5

100

Table 6 Technical parameters of crumb rubber

Item

Unit

Test result

Methods

Density

g/cm3

0.38

GB/T 19208-2008 6.2.4

Ash

%

4.0

GB/T 4498-1997

Acetone extracts

%

12

GB/T 3516-2006

Rubber hydrocarbon content

%

63

GB/T 14837-1993

Fibre content

%

0

GB/T 19208-2008 6.2.3

Metal content

%

0.1

GB/T 19208-2008 6.2.2

Point 10: What about the preparation procedure of the asphalt rubber binder? (e.g. temperature, time, rotation speed, storage stability, etc.)

Response 10: The crumb rubber modified asphalt binder mixing used in this study was the wet process, in which the crumb rubber is added to the virgin asphalt binder (penetration grade 80/100) before introducing it in the asphalt mixture. The crumb rubber modified asphalt binder was produced in the laboratory at 180℃ for 30 min by an open blade mixer at a blending speed of 700 rpm [[1]]. The percentage of crumb rubber added for the crumb rubber modified asphalt binder was 20% by the weight of virgin asphalt. For the warm mix crumb rubber modified asphalt binder, the warm mix additive was added to crumb rubber modified asphalt binder mixing at 180 °C for 30 minutes by a conventional mechanical mixer. Storage stability of binder can be characterized by the difference in softening points, the results are shown in Table 7. Then the crumb rubber modified asphalt binder was prepared for manufacture specimen. A kind of warm asphalt additive (surfactant) was selected in making the warm mix crumb rubber modified asphalt binder. It was added to crumb rubber modified asphalt binder mixing at 180 °C for 30 minutes by a conventional mechanical mixer. Storage stability can be characterized by the difference in softening points. The hot asphalt binder of 50 ± 0.5 g was poured into an aluminum toothpaste tube (32mm in diameter by 160mm in height) in accordance with ASTM D7173 [[2]] then stored vertically in an oven at 163℃, The difference of softening point after 48 hours was shown in table 7.

Table 7 The softening points of the top and bottom of the samples after storing at 163℃ for 48 h

sample

Softening point

bottom

top

Difference

H-CR-60

63.5

57.0

6.5

W-CR-60

60.8

56.2

4.6

H-CR-C

68.1

59.8

8.3

W-CR-C

62.9

57.5

5.4

Point 11: Section 2.2: have you checked if the specimens experienced some damage at the end of the frequency sweeps? Have you verified that 70 microstrain effectively assured the linear viscoelastic domain? Please specify

Response 11: The specimen shall be discarded at the end of any testing series at each temperature period, if the cumulative unrecovered permanent strain was found to be greater than 1500 microstrain, reduce the maximum loading stress level to half. Keep the test data up to this last resting period, discard the specimen, and use a new specimen for the rest of testing periods under reduced load conditions.

AASHTO Standard Test Method TP-15 recommended that a sample be used to assess the stress level required at any given temperature and frequency so that the resulting axial strain is between 75 to 125 microstrain [[3]]. In our study, the highest temperature is only 50°C, which is less than the highest temperature recommended by AASHTO 79-15 (75°C), so the test was specified with the maximum strain not exceeding 70με, which ensured both the test within the linear viscoelastic range and a higher test accuracy. the UTM automatically adjusted the applied load to limit the axial strain of the specimen within 70με.

Point 12: All the symbols introduced in the equations should be explained in the text immediately after the related equations. All the symbols

Response 12: Thank you for your remarkable question. In accordance with your advice, we have checked and carefully revised all of the symbols introduced in the equations in this paper. Ensured that all the symbols introduced in the equations explained in the text immediately after the related equations.

Point 13: The second term of equations 3 and 8 is the same while the first is different. Please check. Similar discrepancies are also presented in some other equations. Please check all again and carefully. And also explain better how the different “sets” of equations are related each other.

Response 13: Thank you for your remarkable question. In accordance with your advice, we have checked and carefully revised all of the equations in this paper. Furthermore, we also have explained better how different “sets” of equations are related each other.

Equation 3 and Equation 8 are generalized sigmoidal models for dynamic modulus and storage modulus, and the model parameters are described in Equations 3 and 8.  The similar problems have been modified.

                                                                                    (3)

Where:  is the logarithm of the dynamic modulus;  is the value of the lower asymptote of the  master curve;  is the difference between the upper and lower asymptotes of the  master curve;  and  is shape coefficients of the  master curve;.  determined the model's asymmetric characteristics.

Equation 8

                                                                                  (8)

Where:  is the logarithm of the storage modulus;  is the value of the lower asymptote of the  master curve;  is the difference between the upper and lower asymptotes of the  master curve;  and  is shape coefficients of the  master curve;.  determined the model's asymmetric characteristics.

Point 14: Line 173: equation 5??

Response 14: equation 6, 7

Point 15: Figure 5: please use a blue curve for WMA-C, like the color of the indicator

Response 15: Thank you for your remarkable question. In accordance with your advice, we have used a blue curve for WMA-C in Figure 5 (in line 608).

Point 16: Lines 362-378: not clear and not thoroughly analyzed

Response 16: Thank you for your remarkable question. In accordance with your advice,

We have revised Lines 362-378 as follows:

The relaxation spectrum peak point is consistent Gundla’s work [[4]]. When relaxation times more than 10-3 s, the discrete relaxation spectrum of HMA-60 is always below HMA-C. That is to say, the discrete relaxation spectrum strength of the former is always smaller than the latter when relaxation times more than 10-3 s. When the relaxation time is less than 10-3 s, the data for former and latter are too few to accurately compare, it is due to the limited of discrete time points. The discrete retardation spectrum of HMA-60 is always above the HMA-C, and the former has a broader retardation spectrum width than the latter. That is to say, the discrete retardation spectrum strength of the former is always higher than the latter. Considering the discrete relaxation spectra and discrete retardation spectra of the two mixtures (HMA-60 and HMA-C), it can be learned that the elastic component of HMA-60 is smaller than HMA-C and the viscous component is larger than HMA-C. Which also indicates that HMA-C only needs a short time to achieve the retardation process under loading. This is because the viscosity of the compound-mesh rubber-modified asphalt binder is greater than the 60-mesh, then the cohesion of HMA-C is also greater than HMA-60, which gives HMA-C greater strength and deformation recovery. The discrete relaxation spectrum of the WMA-60 is always below the WMA-C. That is to say, the discrete relaxation spectrum strength of the former is always smaller than the latter. While the discrete retardation spectrum of WMA-60 is always above the WMA-C, and the former has a broader retardation spectrum width than the latter. That is to say, the discrete retardation spectrum strength of the former is always higher than the latter. Considering the discrete relaxation spectra and discrete retardation spectra of the two mixtures (WMA-60 and WMA-C), it can be learned that the elastic component of WMA-60 is smaller than WMA-C and the viscous component is larger than WMA-C. Which also indicates that WMA-C only needs a short time to achieve the retardation process under loading. This is also because the viscosity of the compound-mesh rubber-modified asphalt binder is greater than the 60-mesh, then the cohesion of WMA-C is also greater than WMA-60, which gives WMA-C greater strength and deformation recovery. Compared the HMCRMA and WMCRMA, once warm mix additive was added, the intensity of the peak relaxation spectrum decreased, the width of the relaxation spectrum decreased, and the discrete relaxation spectrum was difficult to distinguish on the left side of the peak intensity. It means that the elastic component of the mixture is reduced and increased viscous component, so the mixture only needs a shorter time to achieve the relaxation process under load, which is due to the reduced aging effect after the warm mix additive was added.

Point 17: Lines 382-405: justification?

Response 17: Thank you for your remarkable question. In accordance with your advice,

We have revised Lines 382-405 as follows:

The continuous relaxation spectrum is asymmetric, further illustrating the advantages of using the GSM and approximate K-K relationship to fit the master curve of the viscoelastic response function. The magnitude of continuous relaxation spectrum of HMA-60 is less than HMA-C when reduced time greater than 10-3 s and greater than HMA-C once reduced time less than 10-3 s, in other words, the continuous relaxation spectrum strength of the former is less than that of the latter in longer reduced time and greater than the latter in shorter reduced time. However, the magnitude of continuous retardation spectrum of HMA-60 is always greater than the HMA-C in all time domains. Furthermore, the former has a wider retardation spectrum width than the latter. In other words, the strength of the former's continuous retardation spectrum is always greater than the latter in all time domains. Considering the continuous relaxation spectra and continuous retardation spectra of the two mixtures (HMA-60 and HMA-C), it can be learned that the elastic component of HMA-60 is smaller than HMA-C and the viscous component is larger than HMA-C. Which also indicates that HMA-C only needs a short time to achieve the retardation process under loading. This is because the viscosity of the compound-mesh rubber-modified asphalt binder is greater than the 60-mesh, then the cohesion of HMA-C is also greater than HMA-60, which gives HMA-C greater strength and deformation recovery. The magnitude of the continuous relaxation spectrum of the WMA-60 is always less than the WMA-C. In other words, the continuous relaxation spectrum strength of the former is always smaller than that of the latter. While the magnitude of the continuous retardation spectrum of the former is always greater than the latter. That is to say, the strength of the continuous retardation spectrum of the former is always greater than the latter. Considering the continuous relaxation spectra and continuous retardation spectra of the two mixtures (WMA-60 and WMA-C), it can be learned that the elastic component of WMA-60 is smaller than WMA-C and the viscous component is larger than WMA-C. Which also indicates that WMA-C only needs a short time to achieve the retardation process under loading. This is because the viscosity of the compound-mesh rubber-modified asphalt binder is greater than the 60-mesh, then the cohesion of WMA-C is also greater than WMA-60, which gives WMA-C greater strength and deformation recovery. Comparing the HMA and WMA, once warming additive was added, the width of the relaxation (retardation) spectrum narrowed, the peak intensity decreased, and the relaxation (retardation) time corresponding to the peak intensity shifted horizontally to the left. It indicates that the mixture's elastic component is reduced and increased viscous component, so the mixture only needs a shorter time to achieve the relaxation (retardation) process under load. The faster relaxation (retardation) mechanism is due to the reduced aging effect after the warming additives were added. From the molecular point, the primary reason for the above results can be attributed to the decrease in molecular weight and the concentration of polar functional groups in the asphalt binders [[5]].

Point 18: Figures 7 and 8: please use dashed/dotted lines for the continuous spectra

Response 18: Thank you for your remarkable question. In accordance with your advice, we have use dashed lines for the continuous spectra in Figure 7 and 8.

Point 19: Figure 9 joins the data reported in Figures 7 and 8; thus you should use only Figure 9 OR both Figure 7 and 8

Response 19: Thank you for your remarkable question. In accordance with your advice, we have removed Figure 9 from the manuscript retaining Figures 7 and 8.

Point 20: Lines 414-415: ?? the discrete spectra did not seem coincident with the continuous spectra

Response 20: the discrete relaxation and retardation spectra can be obtained by configurations of linear springs and dashpots, respectively. the continuous relaxation and retardation spectra can be obtained by the inverse integral transformation. the continuous spectra can be treated as the limiting case of the discrete ones, in which the time constants are spaced infinitely closely. Figures 7 and 8 are results

Figure 7 The discrete and continuous relaxation spectrum of asphalt mixture

Figure 8 The discrete and continuous retardation spectrum of asphalt mixture

Although discrete and continuous spectra have different peak spectra intensities and peak spectral positions, the shape is the same. For ease understand, the continuous spectrum can be converted to a discrete spectrum, the result was shown in Figure 12.

Figure 12 Compared the strength of discrete and continuous relaxation and retardation spectrum

The discrete spectrum converted from the continuous spectrum is coincident with the discrete spectrum calculated from the Prony series, except for slight differences in the local region, the reasons are as follows

  1. The complex modulus test carried out in temperature of 5℃-50℃, and there are some errors in the test. The higher the temperature, the greater the error.
  2. Using the approximate k-k relations rather than the exact k-k relations. In addition, the GSM constructed from the test results tend to larger errors in extremely low and high frequencies.
  3. In the process of using the Prony series to calculate the discrete spectrum, the discrete time is pre-selected, and the discrete spectrum intensity is obtained by the optimization algorithm, but it is a multi-solution problem, although we set constraints during the optimization process.

Point 21: Lines 416-418: ??

Response 21: These have been revised as follows:

In order to compare discrete and continuous spectra more significantly, the continuous spectrum was converted to its corresponding discrete spectrum by conversion method, and then illustrated it with the discrete spectrum obtained from Prony series in figures 9.

The discrete relaxation spectra can be efficiently calculated from continuous relaxation spectra following equations 39.

                                                                                                      (39)

Where  is discrete relaxation spectrum strength transformed from continuous spectrum;  is continuous spectrum;  is logarithmic interval of discrete relaxation time.

Similarly, the discrete retardation spectra can be efficiently calculated from continuous retardation spectra following equations 40.

                                                                                                        (40)

Where  is discrete retardation spectrum strength transformed from continuous spectrum;  is continuous retardation spectrum;  is logarithmic interval of discrete retardation time.

Figure 9 Compared the strength of discrete spectrum obtained from continuous spectrum and Prony series

Although the shape of the discrete spectrum converted from the continuous spectrum is similar the discrete spectrum calculated from the Prony series, there are some differences in the local region, the reasons are as follows

  1. The complex modulus test carried out in temperature of 5℃-50℃, and there are some errors in the test. The higher the temperature, the greater the error.
  2. Using the approximate k-k relations rather than the exact k-k relations. In addition, the GSM constructed from the test results tend to larger errors in extremely low and high frequencies.
  3. In the process of using the Prony series to calculate the discrete spectrum, the discrete time is pre-selected, and the discrete spectrum intensity is obtained by the optimization algorithm, but it is a multi-solution problem, although we set constraints during the optimization process.

Point 22: Lines 420-422: ??

Response 22: These have been revised as follows:

Point 23: Lines 436-437: what does it mean “the creep test takes a longer time”?

Response 23: Creep is the phenomenon in which constant stress is applied, and the strain gradually increases with time. Longer test times are often required to accurately obtain creep compliance. It is because of the time-consuming process of obtaining creep compliance from creep tests, so creep compliance converted from complex modulus test.

Point 24: Lines 459-472: justification?

Response 24: Compared the relaxation modulus and creep compliance of hot mix crumb rubber modified asphalt mixture, it found that the relaxation modulus of HMA-60 is always less than HMA-C in the entire time domain, which shows the stress relaxation ability of HMA-60 is better than HMA-C. The creep compliance of HMA-60 is higher than HMA-C in the entire time domain, which shows that the deformability of HMA-60 is better than HMA-C. This is because the viscosity of the 60-mesh rubber modified asphalt binder is less than compound-mesh, then the cohesion of HMA-60 is also less than that of HMA-C, which gives HMA-60 a better relaxation and deformation capabilities. Compared the relaxation modulus and creep compliance of warm-mixed crumb rubber modified asphalt mixture, it found that the relaxation modulus of WMA-60 is always less than that of WMA-C in the entire time domain, which shows the stress relaxation ability of WMA-60 is better than WMA-C. The creep compliance of WMA-60 is always higher than the WMA-C in the entire time domain, which shows that the deformability of WMA-60 is better than WMA-C. This is also because the viscosity of the 60-mesh rubber modified asphalt binder is less than compound-mesh, then the cohesion of WMA-60 is also less than that of WMA-C, which gives WMA-60 a better relaxation and deformation capabilities. Comparing the hot-mixed and warm-mixed crumb rubber modified asphalt mixtures, it found that once the warm mix additive was added, the relaxation modulus of crumb rubber-modified asphalt mixtures becomes smaller in the shorter time domain and larger in the longer time domain, creep compliance varies in the opposite way to the relaxation modulus. It shows that the addition of warm mix improves the asphalt mixture's low-temperature stress relaxation ability and high-temperature deformation resistance.

Point 25: Line 505: Figure 14 or 13?

Response 25: Figure. 10

Point 26: Figure 12 also reports the data depicted in Figure 11; thus you should use only Figure 12

Response 26: The Figure 11 allows for a significant comparison of the relaxation modulus and creep compliance of different mixtures. Although the data depicted in Figure 12 are also reported in Figure 11, Figure 12 does not facilitate comparison of the relaxation modulus and creep compliance of different mixtures. Figure 12 is used to compare the differences in relaxation modulus and creep compliance obtained from discrete and continuous spectra.

Point 27: Lines 510-514: this is not showed in Figure 11

Response 27:

Figure 11 The relaxation modulus and creep compliance obtained from the continuous spectrum

It can be learned from Figure. 11: the relaxation modulus and creep compliance obtained from the continuous time spectrum change significantly in the time domain of 10-8 s~107 s, but no significantly change outside the region. When the time domain is greater than 10-5s, the relaxation modulus and creep compliance calculated by the continuous time spectrum are consistent with the discrete time spectrum.

Point 28: Lines 514-518: justification?

Response 28: This is because the magnitude of the continuous relaxation spectrum of HMA-60 is significantly greater than HMA-C in shorter reduced times (lower temperatures), and the higher magnitude of the relaxation spectrum, the higher relaxation modulus.

Point 29: Lines 528-531: justification?

Response 29: The discrete spectra relaxation modulus and creep compliance can be obtained from the discrete time spectrum. The continuous spectra relaxation modulus and creep compliance can be obtained from the continuous time spectrum. For compared, the continuous spectrum can be converted to discrete spectrum and the results are shown in Figure 12.

Figure 12 Compared the strength of discrete and continuous relaxation and retardation spectrum

The discrete relaxation spectra can be efficiently calculated from continuous relaxation spectra following equations 39

                                                                                                       (39)

Where  is discrete relaxation spectrum strength transformed from continuous spectrum;  is continuous spectrum;  is logarithmic interval of discrete relaxation time.

Similarly, the discrete retardation spectra can be efficiently calculated from continuous retardation spectra following equations 40

                                                                                                        (40)

Where  is discrete retardation spectrum strength transformed from continuous spectrum;  is continuous retardation spectrum;  is logarithmic interval of discrete retardation time.

It can be learned from Figure. 12, there is almost no difference between the discrete spectrum and the continuous spectrum in the time region of 10-5 s ~ 103 s.

Point 30: Lines 535-551: justification?

Response 30:

Figure 12 Compared the relaxation modulus and creep compliance obtained from the discrete and continuous spectrum

Point 31: Figure 13: have you already defined the strength?

Response 31: I have defined the continuous spectral intensity of the discrete spectrum as follows:

                                               (15)

                                          (19)

                                              (29)

                                              (37)

Point 32: Figure 14: have you already defined the relative error?

Response 32: Due to an oversight on the part of the authors, to compared the relative errors of the relaxation modulus and creep compliance constructed by discrete and continuous spectrum methods, respectively. The relative errors of the relaxation modulus and creep compliance can be defined follow the Equations. 47 and 48, The result was presented in Figure 12.

The relative error of relaxation modulus

                                                                                                     (47)

Where:  is the relative errors of the relaxation modulus;  is the relaxation modulus calculated from discrete relaxation spectrum;  is the relaxation modulus calculated from continuous relaxation spectrum.

The relative error of creep compliance

                                                                                                    (48)

Where:  is the relative errors of the creep compliance;  is the creep compliance calculated from discrete retardation spectrum;  is the creep compliance calculated from continuous retardation spectrum.

Point 33: Lines 573-574: ???

Response 33: Thank you for your remarkable question. In accordance with your advice, The sentence " The discrete spectrum is coincidence with the continuous spectrum. " in lines 573-574 has been modified to " The shape of the discrete spectrum is coincidence with the continuous spectrum ".

references

[1] Wang H, Li X, Xiao J, et al. High-Temperature Performance and Workability of Crumb Rubber–Modified Warm-Mix Asphalt. Journal of Testing and Evaluation, 2020, 48(4).

[2] ASTM D7173-20, Standard Practice for Determining the Separation Tendency of Polymer from Polymer-Modified Asphalt, ASTM International, West Conshohocken, PA, 2020, www.astm.org

[3] AASHTO T P. Standard method of test for determining the dynamic modulus and flow number for asphalt mixture using the asphalt mixture performance tester. American Association of State Highway and Transportation Officials, Washington, DC, 2015.

[4] Gundla, Akshay, Understanding Viscoelastic Behavior of Asphalt Binders through Molecular Structure Investigation. Arizona State University Phoenix, USA, 2018; ISBN: 9780438297258.

[5] Yu D.; Gu Y.; Yu X. Rheological-microstructural evaluations of the short and long-term aged asphalt binders through relaxation spectra determination. Fuel, 2020, 265: 116953.

Reviewer 3 Report

Please find attached a PDF file with my comments and suggestions for authors.

Author Response

Response to Reviewer 3 Comments

Point 1: First of all, regarding formatting issues, there are several pages with blank parts, due to the arrangement of the figures in the text. It must be corrected once accepted the manuscript. In addition to this, in the figure 9, the picture and its description must be included in the same page

Response 1: Thank you for your remarkable question. In accordance with your advice, we have removed pages with blank parts. In addition, Figure 9 and its description are also shown on the same page. In line 836-840.

Point 2: Regarding the introduction section, it is overall fine and the state‐of‐art is explained with good detail. I suggest to clarify better the aim and the novelty of the study performed. This can be done in the last paragraphs of the introduction section.

Response 2: Thank you for your remarkable question. In accordance with your advice, we have supplemented aim and the novelty of the study performed in the introduction.

This study focused on comparing the relaxation modulus and creep compliance master curves of crumb rubber modified asphalt mixtures constructed by discrete and continuous spectrum methods and to evaluate the accuracy of the master curve model. In line 135-143.

Point 3: In relation to the section “2 Materials and Methods” and “3 Theoretical Background” in my opinion they are fine. I congratulate the authors for the great detail of both sections, especially for the section 3. I think that this great detail is very good because they make easier to understand the work conducted in the research.

Response 3: Thank you for your high praise.

Point 4: In relation to the section “4 Results and Discussion”, I think that the arguments exposed by the authors are adequate. Although, I suggest to include more references in this section, because there are a scarce number of references cited here, and more references are necessary to support the discussion of the results obtained. Despite that, I congratulate the authors for the great detail of the discussion.

Response 4: Thank you for your remarkable question. In accordance with your advice, we have added some references to support the discussion of the results obtained.

Point 5: Regarding the conclusion section, it is generally fine. However, I suggest to authors to consider to summarize the conclusions using bullet points or numbers in order to emphasize the most important findings of the manuscript. This can make the conclusions clearer and perhaps it could be convenient.

Response 5: Thank you for your remarkable question. In accordance with your advice, we have summarized the conclusions using numbers for emphasize the most important findings of the manuscript. The revised conclusions are as follows

In this paper, based on the complex modulus test, we use the discrete and continuous spectrum to construct relaxation modulus and creep compliance master curves. The two methods are consistent with the LVE theory and allow for possible asymmetries in the spectrum curve. The following conclusions can be drawn from this study.

  • According to the complex modulus test data of four types of mixtures, the master curve model of storage modulus and loss modulus is developed according to the approximate K-K relations. These master curve models shared the same model parameters and allowed for possible asymmetry. Similarly, the master curve model of storage compliance and loss compliance is also developed according to the approximate K-K relations.
  • The continuous and discrete relaxation spectrum can be obtained from the Storage modulus master curves utilizing integral transformations and the collocation method, respectively. Similarly, the continuous and discrete retardation spectrum can be obtained from the storage compliance master curve.
  • The shape of the discrete spectrum is similar to the continuous spectrum. The most significant difference is the intensity of the discrete spectrum is higher than of the continuous spectrum.
  • The relaxation modulus and creep compliance were calculated by the discrete and continuous spectrum, respectively. The difference between the discrete and continuous spectrum is negligible in the time domain of 10-5 s-103 s. However, significant differences outside the region.
  • HMA-60 has a better deformation capacity than HMA-C at shorter reduced time and a worse deformation resistance than HMA-C at longer reduced time. Moreover, the deformation capacity of WMA-60 is better than WMA-C at shorter reduced time, and a worse deformation resistance than WMA-C at longer reduced time. Once the warm mix additive was added, the mixture's deformation capacity was improved at a shorter reduced time. Moreover, improved the deformation resistance in longer reduced time.

Finally, despite the comments, I want to encourage the authors for continuing working in this research topic, and I think that the manuscript could be published in Materials journal, after including the proposed changes.

Reviewer 4 Report

Introduction

The authors did not show the novelty and aim of this work. In addition, the authors can not prove the novelty of this work without a recent literature. Furthermore, most of the cited study are from Asia. Please, check studies in other continents (Europe and America).

Materials and Methods

Please, add a paragraph regarding the next sub-sections. Show why and how they are divided.

Results and Discussion

The major issue regarding this work can be seen in this section. The authors simply showed the results without giving any proper discussion. Thus, they must compare their works with the previous studies.

Also, the size of the texts in the most figures are very small. The reviewer could not understand them. Thus, the reviewer may give more comments in the next revision.

Conclusion

It is well written.

Author Response

Response to Reviewer 4 Comments

Introduction

Point 1: The authors did not show the novelty and aim of this work. In addition, the authors can’t prove the novelty of this work without a recent literature. Furthermore, most of the cited study are from Asia. Please, check studies in other continents (Europe and America).

Response 1: Thank you for your remarkable question. In accordance with your advice, we have added the novelty and aim of this work (in line 135-137). In addition, recent literature is also added (references 26-34), but in recent years research on this subject has focused more on its applications, especially in Europe and the United States. This study focused on comparing the relaxation modulus and creep compliance master curves of crumb rubber modified asphalt mixtures constructed by discrete and continuous spectrum methods and to evaluate the accuracy of the master curve model.

Materials and Methods

Point 2: Please, add a paragraph regarding the next sub-sections. Show why and how they are divided.

Response 2: Thank you for your remarkable question. In accordance with your advice, we have added a paragraph to distinguish the different of types mixtures. (in line 162-170). The details are as follows

Hot mix 60-mesh crumb rubber modified asphalt mixture (HMA-60) is the mixture prepared by mixing aggregates and 60-mesh crumb rubber modified asphalt binder following the hot mixing process. While warm mix 60-mesh crumb rubber modified asphalt mixture (WMA-60) is the mixture prepared by mixing aggregate and warm mix 60-mesh crumb rubber modified asphalt binder following the warm mixing process. Hot mix compound-mesh crumb rubber modified asphalt mixture (HMA-C) is the mixture prepared by mixing aggregates and compound-mesh crumb rubber modified asphalt binder following the hot mixing process. Warm mix compound-mesh crumb rubber modified asphalt mixture (WMA-C) is the mixture prepared by mixing aggregates and warm mix compound-mesh crumb rubber modified asphalt binder following the warm mixing process.

Table 1 production temperatures and components for types of mixture

Item

Mixed Temperature

Compacted

Temperature

Type of crumb rubber for binder

Whether or not the warm mix additive was added

HMA-60

180℃

170℃

60-mesh

No

WMA-60

162℃

152℃

60-mesh

Yes

HMA-C

180℃

170℃

Compounded-mesh

No

WMA-C

162℃

152℃

Compounded-mesh

Yes

Results and Discussion

Point 3: The major issue regarding this work can be seen in this section. The authors simply showed the results without giving any proper discussion. Thus, they must compare their works with the previous studies.

Response 3: Thank you for your remarkable question. In accordance with your advice,

We have compared our works with previous studies and given proper discussion. (in line 612-614, 758-760, 877-882)

Also, the size of the texts in the most figures are very small. The reviewer could not understand them. Thus, the reviewer may give more comments in the next revision.

Response 3: Thank you for your remarkable question. In accordance with your advice, we have expanded the texts in the figures.

Conclusion

It is well written.

Reviewer 5 Report

The paper discussion is clear and convincing. The quality of pictures are satisfying. The following comments are added:

1) The authors mentioned that outside time range is 55% error. What is the source of such uncertainty? 

I recommend that the authors write one or two lines about that. Furthermore, the authors ahve any comment to minimise the uncertainty?

2) How much WMA additive was used?

3) Table 2 only shows properties of HMA. It is recommended to shows properties of WMA.

Author Response

Response to Reviewer 5 Comments

The paper discussion is clear and convincing. The quality of pictures are satisfying. The following comments are added:

Point 1: The authors mentioned that outside time range is 55% error. What is the source of such uncertainty?

Response 1: To compare the relative errors of the relaxation modulus and creep compliance constructed by discrete and continuous spectrum methods, respectively. The relative errors of the relaxation modulus and creep compliance can be defined follow the Equations. 45 and 46, The result was presented in Figure 13.

The relative error of relaxation modulus

                                                                                                     (47)

Where:  is the relative errors of the relaxation modulus;  is the relaxation modulus calculated from discrete relaxation spectrum;  is the relaxation modulus calculated from continuous relaxation spectrum.

The relative error of creep compliance

                                                                                                    (48)

Where:  is the relative errors of the creep compliance;  is the creep compliance calculated from discrete retardation spectrum;  is the creep compliance calculated from continuous retardation spectrum.  

Figure 13 Relative errors of the relaxation modulus and creep compliance produced by methods of the discrete and continuous spectrum

outside time range of 10-5 s~103 s, the maximum relative errors for relaxation modulus and creep compliance is 39% and 55%, respectively. This is because the discrete and continuous spectra are also quite different outside this region. In order to compare discrete and continuous spectra more significantly, the continuous spectrum was converted to its corresponding discrete spectrum by conversion method, and then illustrated it with the discrete spectrum obtained from Prony series in figures 9.

The discrete relaxation spectra can be efficiently calculated from continuous relaxation spectra following equations 39.

                                                                                                      (39)

Where  is discrete relaxation spectrum strength transformed from continuous spectrum;  is continuous spectrum;  is logarithmic interval of discrete relaxation time.

Similarly, the discrete retardation spectra can be efficiently calculated from continuous retardation spectra following equations 40.

                                                                                                        (40)

Where  is discrete retardation spectrum strength transformed from continuous spectrum;  is continuous retardation spectrum;  is logarithmic interval of discrete retardation time.

Figure 9 Compared the strength of discrete spectrum obtained from continuous spectrum and Prony series

As shown in Figure 9. It can be seen from the figure that little difference within the region and significant difference outside the region

The reasons for the uncertainty are as follows:

  1. The complex modulus test carried out in temperature of 5℃-50℃, and there are some errors in the test. The higher the temperature, the greater the error.
  2. Using the approximate k-k relations rather than the exact k-k relations. In addition, the GSM constructed from the test results tend to larger errors in extremely low and high frequencies.
  3. In the process of using the Prony series to calculate the discrete spectrum, the discrete time is pre-selected, and the discrete spectrum intensity is obtained by the optimization algorithm, but it is a multi-solution problem, although we set constraints during the optimization process.

I recommend that the authors write one or two lines about that. Furthermore, the authors have any comment to minimise the uncertainty?

Response: Taking all factors into account, I believe that the following measures can be taken to minimise the uncertainty

  1. Perform the complex modulus tests at a wider range of temperatures and increase the number of test specimens especially in high temperature.
  2. The data can be checked in Black Space before the master curve model is fitted. Moreover, the constrained upper shelf of the master curve should be obtained from other methods.
  3. In the process of calculating the discrete spectrum with the Prony series, the discrete time is pre-selected. The intensity of the discrete spectrum is obtained through the optimization algorithm, and the discrete time is adjusted gradually in the optimization process, so as to achieve the simultaneously optimization of the discrete time and the intensity of spectrum.

Point 2: How much WMA additive was used?

Response 2: The content of warm mix additive is 1% by the weight of asphalt binder.

Point 3: Table 2 only shows properties of HMA. It is recommended to shows properties of WMA.

Response 3: Thank you for your remarkable question. In accordance with your advice, we have added properties of WMA in Table 3.

Table 3 Volume parameters of optimum asphalt content

Asphalt

Mixture

OAC/%

Gross density g/cm³

Theoretical density g/cm³

Void/ %

VMA/%

VFA/%

Stability/KN

Flow value/

mm

HMA-60

5.4

2.439

2.537

3.86

14.04

72.5

10.45

2.74

WMA-60

5.4

2.442

2.538

3.78

13.93

72.8

10.95

2.87

HMA-C

5.6

2.441

2.546

4.12

14.15

70.9

11.23

2.45

WMA-C

5.6

2.444

2.546

4.01

14.04

71.5

11.61

2.62

Standard

3~5

≧13.

65~75

≧8

2~4

Round 2

Reviewer 1 Report

The authors took into account the comments of the reviewer.

In general, the article met the acceptance requirements.

Reviewer 2 Report

I am still afraid the paper could represent a pure mathematical exercise without relevant practical implications for researchers and applicants.

However, the quality of the paper has been significantly improved and it could be published in the present form if the other reviewrs also agree 

Reviewer 4 Report

The manuscript can be now recommended for publication.

Please, check the order of the tables (e.g. Table 3 came before the text)